

# Lattice simulations of non-minimally coupled scalar fields in the Jordan frame

**Daniel G. Figueroa[1*], Adrien Florio[2†], Toby Opferkuch[3,4,5‡] and Ben A. Stefanek[6∘]**

**1** Instituto de Física Corpuscular (IFIC), Universitat de València-CSIC,
E-46980, Valencia, Spain
**2** Center for Nuclear Theory, Department of Physics and Astronomy,
Stony Brook University Stony Brook, New York 11794, USA
**3** Berkeley Center for Theoretical Physics, University of California, Berkeley, CA 94720, USA
**4** Theoretical Physics Group, Lawrence Berkeley National Laboratory,
Berkeley, CA 94720, USA
**5** Theoretical Physics Department, CERN, Esplanade des Particules,
1211 Geneva 23, Switzerland
**6** Physik-Institut, Universität Zürich, CH-8057 Zürich, Switzerland

⋆ daniel.figueroa@ific.uv.es , † adrien.florio@stonybrook.edu ,
‡ toby.opferkuch@cern.ch , ∘ bestef@physik.uzh.ch

## Abstract

The presence of scalar fields with non-minimal gravitational interactions of the form $\xi|\phi|^2 R$ may have important implications for the physics of the early universe. We propose a procedure to solve the dynamics of non-minimally coupled scalar fields directly in the Jordan frame, where the non-minimal couplings are maintained explicitly. Our algorithm can be applied to lattice simulations that include minimally coupled fields and an arbitrary number of non-minimally coupled scalars, with the expansion of the universe sourced by all fields present. This includes situations when the dynamics become fully inhomogeneous, fully non-linear (due to e.g. backreaction or mode rescattering effects), and/or when the expansion of the universe is dominated by non-minimally coupled species. As an example, we study geometric preheating with a non-minimally coupled scalar spectator field when the inflaton oscillates following the end of inflation.

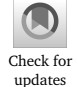

## Contents



# 1   Introduction

The dynamics of fields with non-minimal gravitational interactions may have important implications for the physics of the early universe. In the case of scalar field $\phi$ (either singlet or charged), one can add to the action an operator of the form $\xi|\phi|^2 R$, where $R$ is the Ricci scalar and $\xi$ is a real coupling constant controlling the strength of the interaction. The presence of such term is actually required by the renormalization properties of a scalar field in curved spacetime [1, 2], where it is a running parameter that cannot be set to zero at all energy scales.[1] In the case of the Standard Model (SM) Higgs field $\Phi$, the operator $\xi|\Phi|^2 R$ is actually the only missing operator of dimension-4 that respects all symmetries of the SM and gravity. The coupling $\xi$ can therefore be considered as the last unknown parameter of the SM. However, due to the weakness of the gravitational interaction, current particle physics experiments provide only extremely weak constraints on this coupling, $\xi \lesssim 10^{15}$ [3]. It is therefore likely that only early universe phenomena involving much higher energies than those accessible to particle colliders can allow us to probe the non-minimal gravitational interaction of the SM Higgs field, see e.g. [4–13].

Other fundamental (yet speculative) scalar fields may also have non-minimal interactions with gravity. For instance, an early phase of accelerated expansion in the Universe, known as *inflation*, is often assumed to be driven by a scalar field called the *inflaton*, with an appropriate potential and initial conditions (for reviews on inflation see e.g. [14–18]). Indeed, a scalar field with a non-minimal coupling to gravity can actually serve as a good inflaton candidate, as any non-minimally coupled scalar theory can be mapped via a conformal transformation to a minimally coupled theory with an effective potential that can sustain inflation. Another popular realization of inflation lies in *modified gravity* $f(R)$ theories, where $f$ is an arbitrary function of $R$ (for a review on $f(R)$ theories, see e.g. [19]). If $f'(R) \neq 0$ and $f''(R) \neq 0$, there always exists a mapping between the $f(R)$ theory and a scalar-tensor theory with a propagating scalar degree of freedom non-minimally coupled to gravity with a scalar potential purely of gravitational origin. As previously mentioned, this setup can then be mapped onto a minimally coupled theory with an effective potential suitable for inflation. A paradigmatic example of this is *Starobinsky inflation* [20], defined by $f(R) = R + \alpha R^2$ with $\alpha > 0$. After a conformal transformation to obtain a minimally coupled theory, there is a scalar field – the *scalaron* – with a potential that plateaus at large field amplitudes, naturally leading to

---

[1]Exceptionally, the running vanishes for the conformal value $\xi = 1/6$.

inflation. Scenarios where the inflaton has a non-minimal gravitational coupling $\xi|\phi|^2 R$ lead to inflationary predictions in excellent agreement with current observational constraints [21, 22]. This is independent of whether the inflaton is of gravitational origin (as in Starobinsky inflation) or elementary origin as in *Higgs-Inflation*, where the inflaton is identified with the SM Higgs [23]. It is very interesting that data from cosmological observations clearly favours plateau-like potentials that naturally emerge in these scenarios [21].

Non-minimally coupled inflaton scenarios can also lead to very interesting phenomenology during the period after inflation. If the inflaton oscillates around the minimum of its potential following inflation, particle species coupled with sufficient strength are typically created in energetic bursts. This non-perturbative process of particle production is known as *preheating*, and it often leads to an exponential transfer of energy into particle sectors (for reviews see [24–27]). This can occur in non-minimally coupled inflaton $\chi$ scenarios, where preheating into other degrees of freedom, e.g. scalar fields $\{\phi\}$, can be realized very efficiently when couplings of the form $g^2\chi^2\phi^2$ or $\xi\phi^2 R$ are considered. Preheating scenarios considering a simple monomial inflaton potential and a preheat scalar field non-minimally coupled to gravity $\xi\phi^2 R$ were first considered in [28] and later on in [29], where an inflaton-preheat field coupling $g^2\chi^2\phi^2$ was also included. The excitation of the non-minimally coupled preheat field due to the oscillatory behavior of the curvature $R$ (dictated by the oscillations of the inflaton field) was coined as *Geometric preheating* in Ref. [28], and we keep that terminology here. Preheating following inflation due to higher order curvature terms $f(R) = R + \alpha_n R^n$ has been studied considering geometric preheating effects in [30, 31]. Preheating after Higgs-Inflation was originally studied in [32–34], considered in more detail in [7, 35], and lastly for modified setups, like $R^2$-Higgs inflation, in [36]. Preheating in multi-field inflationary scenarios with $N$ scalar fields $\{\phi_j\}$ and couplings $\xi_j\phi_j^2 R$ has also been extensively studied [37–41]. Finally, preheating with a non-minimal gravitational coupling $f(\phi)R$ with $f$ a general function of $\phi$, has been also considered, see e.g. [42]. All of the above instabilities due to the presence of non-minimal gravitational couplings can be generically regarded as "gravitational reheating" mechanisms,[2] as they lead to very efficient preheating, often exhibiting a violent transfer of energy among fields. A different type of gravitational reheating mechanism was originally put forward in [43] (see also [44]). Namely, a massless scalar field $\phi$ non-minimally coupled to gravity is excited towards the end of inflation.[3] The inflaton potential is chosen such that there is a sustained *kination* dominated era with stiff equation of state $1/3 < w \leq 1$ after inflation. As a consequence, the energy stored in $\phi$ (initially suppressed compared to the inflaton energy) eventually becomes the dominant energy component of the Universe. This idea is perhaps best exemplified in so-called *Quintessential inflation* [46–54]. The original gravitational reheating mechanism, however, was shown in Ref. [55] to be inconsistent with BBN/CMB constraints [56, 57] due to an excess amount of gravitational wave production.[4] More relevantly, it has been also shown that a massless spectator field with a non-minimal coupling to gravity does actually not scale as a radiation degree of freedom during kination domination (as originally assumed in [43, 44, 46]), but rather experiences a tachyonic instability due to the change in sign of $R \propto (1-3w)$ for a stiff equation of state $w > 1/3$. If the field is also self-interacting, its energy grows due to the tachyonic instability until the self-interaction eventually compensates the tachyonic mass. This was first considered in Ref. [63], with the SM Higgs as a spectator field with a non-minimal coupling to gravity. There the universe is

---

[2]This terminology does not apply to scenarios like Higgs-inflation, where only the inflaton is non-minimally coupled to gravity, while the preheat fields are directly coupled (non-gravitationally) to the inflaton.

[3]In [43] a non-conformal coupling was considered, i.e. $\xi \neq 1/6$, but assuming a quasi-conformal window $0 < |6\xi - 1| \ll 1$. Ref. [45] showed later on that the $(6\xi - 1)^2$ suppression of the energy density there is lifted to $\mathcal{O}(1)$ away from the quasi-conformal window.

[4]In these scenarios the gravitational wave background from inflation develops a large blue-tilt at high frequencies, see e.g. [58–62].

reheated into relativistic SM particles after the Higgs experiences tachyonic growth during a period of kination domination and then decays. The same mechanism was later studied in more detail and extended to generic scalar fields non-minimally coupled to gravity in [61], see also [64, 65]. In [61], this mechanism was coined as *Ricci reheating* and we stick to that nomenclature here.

All of the above scenarios exemplify the relevance of understanding the dynamics of scalar fields non-minimally coupled to gravity in the early universe. The form of the action where the non-minimal coupling to gravity is maintained explicitly is known as the *Jordan frame*. In this frame, the resulting equations of motion are difficult to solve in full generality due to the non-linear feedback among them. Consequently, most studies rely on a conformal transformation of the metric that brings the gravitational action to the canonical Einstein-Hilbert form. This defines the so-called *Einstein frame*, where the non-minimal coupling is absent and instead the kinetic terms and scalar potentials of the matter fields are multiplied by a conformal factor depending on the non-minimal coupling. Most of the studies cited above have worked out the dynamics of non-minimally coupled scalar fields in the Einstein frame, or in the linear regime in the Jordan frame, where analytic calculations can be employed. The two frames are equivalent at the classical level, as long as the map between them is non-singular. However, explicit examples exist where the conformal map does not exist for all field values, such as in the transformation from a non-minimally coupled theory to a minimally coupled theory. In this case, the conformal map is given by $\Omega^2 = 1 - \xi(\phi/m_p)^2$ which appears to be non-invertible for $\phi^2 = m_p^2/\xi$ and $\xi > 0$ (here $m_p \simeq 2.4 \times 10^{18}$ GeV is the *reduced Planck Mass*). Furthermore, it is not known to what extent the two frames are equivalent at the quantum level, as the conformal factor to change from the Jordan to the Einstein frame is a local function of the non-minimally coupled field $\Omega^2(\phi(x))$, with $\phi(x)$ often treated as a quantum field. Some works evaluate the map using the vacuum expectation value $\langle \phi^2 \rangle$, but then it is not clear that the Einstein frame description fully captures the physics of the theory originally written in the Jordan frame, especially in the case where the initial conditions are determined purely through quantum fluctuations.

In this paper, we introduce a technique for solving the system directly as written in the original Jordan frame, avoiding the need to perform any conformal transformation. In particular, we are able to solve the dynamics of an arbitrary scalar field $\phi$ with a non-minimal coupling to gravity $\xi\phi^2 R$ in an expanding background sourced by all fields present, even when the dynamics become fully inhomogeneous and/or fully non-linear due to backreaction of the excited species, including when the expansion of the universe is dominated by the non-minimally coupled species. We can self-consistently evolve the expansion of the universe while fully capturing field inhomogeneities and non-linearities in the system, both of which typically develop very rapidly when there are exponential instabilities like those typically arising in the presence of a non-minimally coupled scalar field. As a working example, we study geometric preheating effects involving a real scalar spectator field non-minimally coupled to gravity, excited via an oscillatory effective mass from $R$ that is sourced by oscillations of an inflaton with monomial potential around its minimum.

## 2 Continuum dynamics in the Jordan frame

In this section we derive the equations of motion in the Jordan frame for a theory with a non-minimally coupled scalar field. We consider a flat Friedmann-Lemaître-Robertson-Walker (FLRW) background described by

$$ds^2 = -a(\eta)^{2\alpha}d\eta^2 + a(\eta)^2\delta_{ij}dx^i dx^j, \tag{1}$$

with $\eta$ an "$\alpha$-time" variable related to cosmic time by $dt = a(\eta)^\alpha d\eta$. Here $\alpha$ is a (real number) parameter to be conveniently chosen to suit each particular problem. Given the metric in Eq. (1), the Ricci scalar $R$ can be computed as (see Appendix A)

$$R = \frac{6}{a^{2\alpha}} \left[ \frac{a''}{a} + (1-\alpha)\left(\frac{a'}{a}\right)^2 \right],\tag{2}$$

where primes indicate derivatives with respect to $\eta$. We emphasize that the Ricci scalar is a spatially homogeneous function, only depending on time, as expected from consistency with Eq. (1). Let us consider a generic matter sector $\{\varphi_m\}$ minimally coupled to gravity, together with a scalar field $\phi$ non-minimally coupled to gravity. Without loss of generality, the action of this system reads

$$\mathcal{S} = \int d^4x \sqrt{-g} \left( \frac{1}{2}m_p^2 R - \frac{1}{2}\xi R\phi^2 - \frac{1}{2}g^{\mu\nu}\partial_\mu\phi\,\partial_\nu\phi - V(\phi, \{\varphi_m\}) + \mathcal{L}_m \right),\tag{3}$$

where $\frac{1}{2}m_p^2 R$ is the standard *Hibert-Einstein* term, $\frac{1}{2}\xi R\phi^2$ represents a non-minimal gravitational interaction of $\phi$, and $V(\phi, \{\varphi_m\})$ encompasses both the self interactions of $\phi$ as well as its non-gravitational interactions with the minimally-coupled matter sector. The term $\mathcal{L}_m$ characterizes the dynamics of the minimally-coupled fields, including their interactions and self-interactions (which we do not specify explicitly here since they are irrelevant for our discussion). Varying the action with respect to $\phi$, we obtain the following equation of motion for $\phi$

$$\Box\phi - \xi R\phi - \frac{\partial V}{\partial \phi} = 0,\tag{4}$$

where $\Box = g^{\mu\nu}\nabla_\mu\nabla_\nu$ and $\nabla_\mu$ is the covariant derivative. We see that the non-minimal coupling introduces a term proportional to $R$ in the equation of motion that acts a time dependent effective mass for $\phi$. Using the $\alpha$-time metric given in Eq. (1), the above equation becomes

$$\phi'' + (3-\alpha)\frac{a'}{a}\phi' - a^{-2(1-\alpha)}\nabla^2\phi + a^{2\alpha}\left(\xi R\phi + \frac{\partial V}{\partial \phi}\right) = 0.\tag{5}$$

Equivalently, we can think of this gravitational interaction as part of an effective potential

$$V_{\text{eff}}(\phi, \{\varphi_m\}, R) \equiv V(\phi, \{\varphi_m\}) + \frac{1}{2}\xi R\phi^2,\tag{6}$$

that includes all together the non-minimal coupling to gravity, the non-gravitational interactions with the minimally-coupled matter sector, as well as the self-interactions of $\phi$. For convenience, we can then think of a Lagrangian for $\phi$ given by

$$\mathcal{L}_\phi \equiv -\frac{1}{2}g^{\mu\nu}\partial_\mu\phi\,\partial_\nu\phi - V_{\text{eff}}(\phi, \{\varphi_m\}, R).\tag{7}$$

The Einstein equations are obtained by varying Eq. (3) with respect to $g^{\mu\nu}$

$$G_{\mu\nu} = R_{\mu\nu} - \frac{1}{2}g_{\mu\nu}R = \frac{1}{m_p^2}T_{\mu\nu},\tag{8}$$

with

$$T_{\mu\nu} = -\frac{2}{\sqrt{-g}}\frac{\delta(\sqrt{-g}\mathcal{L}_m)}{\delta g^{\mu\nu}} - \frac{2}{\sqrt{-g}}\frac{\delta(\sqrt{-g}\mathcal{L}_\phi)}{\delta g^{\mu\nu}} \equiv T_{\mu\nu}^m + T_{\mu\nu}^\phi,\tag{9}$$

where $T_{\mu\nu}^{\mathrm{m}}$ and $T_{\mu\nu}^{\phi}$ have been defined as the energy-momentum tensors of the minimally-coupled matter fields and the non-minimally coupled scalar field $\phi$, respectively. In particular, one finds (see Appendix B) the energy-momentum tensor of the non-minimally coupled field to be

$$T_{\mu\nu}^{\phi} = \partial_\mu\phi\,\partial_\nu\phi - g_{\mu\nu}\left(\frac{1}{2}g^{\rho\sigma}\partial_\rho\phi\,\partial_\sigma\phi + V\right) + \xi(G_{\mu\nu} + g_{\mu\nu}\Box - \nabla_\mu\nabla_\nu)\phi^2\,. \tag{10}$$

The trace of $T_{\mu\nu}^{\phi}$, defined as $T_\phi = g^{\mu\nu}T_{\mu\nu}^{\phi}$, takes a simple form and will prove very useful for simplifying the equations determining the evolution of the scale factor. In $d+1$ spacetime dimensions, we find

$$T_\phi = (1-d)\frac{1}{2}\partial^\mu\phi\,\partial_\mu\phi - (d+1)V + \xi G\phi^2 + d\,\xi\Box\phi^2\,, \tag{11}$$

where $G = g^{\mu\nu}G_{\mu\nu} = (1-d)R/2$ is the trace of the Einstein tensor with respect to the background metric. Taking $d = 3$ and using Eq. (4), we obtain

$$T_\phi = (6\xi - 1)\left(\partial^\mu\phi\,\partial_\mu\phi + \xi R\phi^2\right) + 6\xi\phi V_{,\phi} - 4V\,, \tag{12}$$

where $V_{,\phi} = \partial V/\partial\phi$. Notably, if $\xi = 1/6$, then for $V = 0$ or $V \propto \phi^4$ we find that $T_{\mu\nu}^{\phi}$ is traceless, i.e. $T_\phi = 0$, as a consequence of the conformal invariance of $\mathcal{S}_\phi = \int d^4x\sqrt{-g}\mathcal{L}_\phi$ in these cases. Given the FLRW metric in Eq. (1), the consistency of the Einstein equations requires that $T_{\mu\nu}$ takes the form of the energy-momentum tensor of a perfect fluid $T^\mu_{\ \nu} = \mathrm{diag}\{-\rho(\eta), p(\eta), p(\eta), p(\eta)\}$. We note that while fields can develop large spatial inhomogeneities, the homogeneous and isotropic pressure and energy density $p(\eta)$ and $\rho(\eta)$ should be understood as the result of a volume average over the inhomogeneous local field expressions. When the averaging volume is sufficiently large compared to the excitation scales of the fields, this procedure leads to a well-defined notion of a homogeneous and isotropic pressure and energy density within the given volume. In this case, taking spatial averages over the off-diagonal elements of $T_{\mu\nu}$ leads to vanishing results, consistent with homogeneity and isotropy within the considered volume. Under these conditions, the Einstein equations reduce to the Friedmann equations in $\alpha$-time

$$\mathcal{H}^2 \equiv \left(\frac{a'}{a}\right)^2 = \frac{a^{2\alpha}}{3m_p^2}\rho(\eta)\,, \tag{13}$$

$$\frac{a''}{a} = -\frac{a^{2\alpha}}{6m_p^2}\Big[(1-2\alpha)\rho(\eta) + 3p(\eta)\Big]\,, \tag{14}$$

where we defined $\mathcal{H} = a'/a$, which is related to the cosmic time Hubble rate $H$ as $H = \mathcal{H}/a^\alpha$. We define the energy density and pressure as

$$\rho(\eta) = \rho_\phi(\eta) + \rho_{\mathrm{m}}(\eta) \equiv a^{-2\alpha}\langle T_{00}^{\phi}\rangle + a^{-2\alpha}\langle T_{00}^{\mathrm{m}}\rangle\,, \tag{15}$$

$$p(\eta) = p_\phi(\eta) + p_{\mathrm{m}}(\eta) \equiv \frac{1}{3a^2}\delta^{ij}\langle T_{ij}^{\phi}\rangle + \frac{1}{3a^2}\delta^{ij}\langle T_{ij}^{\mathrm{m}}\rangle\,, \tag{16}$$

with $\langle\ldots\rangle$ denoting volume averages. With these definitions, the explicit expressions for the energy density and pressure of the non-minimally coupled field $\rho_\phi$ and $p_\phi$ are found to be[5]

---

[5]The volume averages of the total divergence terms $\nabla^2\phi^2 = \nabla\cdot\nabla\phi^2$ can be converted into surface integrals that vanish in the case of an infinite volume with well-behaved fields or in the case of a finite volume with periodic boundary conditions.

$$\rho_\phi(\eta) = \frac{1}{2a^{2\alpha}}\langle\phi'^2\rangle + \frac{1}{2a^2}\langle(\nabla\phi)^2\rangle + \langle V(\phi)\rangle + \frac{3\xi}{a^{2\alpha}}\mathcal{H}^2\langle\phi^2\rangle + \frac{6\xi}{a^{2\alpha}}\mathcal{H}\langle\phi\phi'\rangle - \frac{\xi}{a^2}\langle\nabla^2\phi^2\rangle,$$
$$(17)$$

$$p_\phi(\eta) = \frac{(1-4\xi)}{2a^{2\alpha}}\langle\phi'^2\rangle - \frac{(1-12\xi)}{6a^2}\langle(\nabla\phi)^2\rangle - \langle V(\phi)\rangle + \frac{2\xi}{a^{2\alpha}}\mathcal{H}\langle\phi\phi'\rangle - \frac{\xi}{3a^2}\langle\nabla^2\phi^2\rangle$$
$$+ 2\xi\langle\phi V_{,\phi}\rangle + \frac{\xi}{a^{2\alpha}}\left[\mathcal{H}^2 + 12\left(\xi - \frac{1}{6}\right)\left(\frac{a''}{a} + (1-\alpha)\mathcal{H}^2\right)\right]\langle\phi^2\rangle. \qquad (18)$$

In principle, one can solve for the scale factor $a(\eta)$ from either Eq. (13) or Eq. (14). However, it is difficult in practice to solve these equations due to their non-linear dependence on the derivatives of the scale factor. An alternative approach is to relate the evolution of the scale factor to the trace of the energy-momentum tensor, which only includes terms involving $R$ and the fields. An expedient way to do this is by computing the trace of Eq. (8), which gives

$$R = -\frac{1}{m_p^2}g^{\mu\nu}\left(T_{\mu\nu}^\phi + T_{\mu\nu}^{\mathrm{m}}\right) = -\frac{1}{m_p^2}\left(T_\phi + T_{\mathrm{m}}\right). \qquad (19)$$

Inserting the expression for $T_\phi$ given in Eq. (12), taking the volume average of both sides, and solving for $R$, we find an expression only in terms of the fields

$$R = \frac{F(\phi)}{m_p^2}\Big[(1-6\xi)\langle\partial^\mu\phi\partial_\mu\phi\rangle + 4\langle V\rangle - 6\xi\langle\phi V_{,\phi}\rangle - \langle T_{\mathrm{m}}\rangle\Big],$$
$$F(\phi) \equiv \frac{1}{1 + (6\xi-1)\xi\langle\phi^2\rangle/m_p^2}. \qquad (20)$$

This expression for $R$ can be directly related to the evolution of the scale factor using Eq. (2). This leads to the differential equation[6]

$$\frac{a''}{a} + (1-\alpha)\left(\frac{a'}{a}\right)^2 = \frac{a^{2\alpha}F(\phi)}{6m_p^2}\Big[(1-6\xi)\langle\partial^\mu\phi\partial_\mu\phi\rangle + 4\langle V\rangle - 6\xi\langle\phi V_{,\phi}\rangle - \langle T_{\mathrm{m}}\rangle\Big], \qquad (21)$$

that together with the equation of motion for $\phi$ in Eq. (5), will allow us to spell out a simple and concise numerical scheme to evolve this system. To start, it is convenient write the equations in terms of *natural variables*, by rescaling fields and coordinates as

$$\tilde{\phi} = \frac{1}{f_*}\phi, \quad \mathrm{d}\tilde{\eta} = \omega_*\mathrm{d}\eta, \quad \mathrm{d}\tilde{x}_i = \omega_*\mathrm{d}x_i. \qquad (22)$$

with $f_*$ some typical field amplitude and $\omega_*$ a characteristic (inverse) time scale of the problem to be studied. The choice of $f_*$ and $\omega_*$ depends entirely on the scenario at hand (we will provide an explicit example in Section 4. We also need introduce an appropriate rescaling of the matter sector (see Ref. [67] for examples). If the matter sector simply comprises of a set of scalar fields $\{\varphi_{\mathrm{m}}\}$, these are normalized as in Eq. (22). We note that rescaling the coordinates by $\omega_*$ naturally induces the following rescaling in $R$

$$\tilde{R} = \omega_*^{-2}R. \qquad (23)$$

It is also natural to introduce rescaled energy densities and pressure

$$\tilde{V} = \frac{1}{f_*^2\omega_*^2}V, \quad \tilde{\rho} = \frac{1}{f_*^2\omega_*^2}\rho, \quad \tilde{p} = \frac{1}{f_*^2\omega_*^2}p. \qquad (24)$$

---

[6]We note that this matches Eq. 12 of Ref. [66] in the case of a quartic potential and $\alpha = 1$ (corresponding to conformal time).

Next, we reduce the order of the equation of motion for $\tilde{\phi}$ by introducing a conjugate momentum variable as

$$\tilde{\pi}_\phi = a^{3-\alpha}\tilde{\phi}'. \tag{25}$$

The matter sector is treated in a similar way, with the rescaling of the conjugate momenta variables depending on the spin of the species, see [67]. If the matter sector is comprised of scalar fields, we simply introduce a set of conjugate momenta $\{\tilde{\pi}_{\varphi_m}\}$, analogously to Eq. (25). In the new variables, the evolution of the non-minimally coupled scalar field is governed by a system of coupled first-order differential equations, in terms of a *kernel* functional $\tilde{\mathcal{K}}_\phi$, as follows

$$\begin{cases} \tilde{\phi}' = a^{\alpha-3}\tilde{\pi}_\phi\,, \\ \tilde{\pi}'_\phi = \tilde{\mathcal{K}}_\phi[a,\tilde{\phi},\{\tilde{\varphi}_m\},\tilde{R}], \quad \text{with} \quad \mathcal{K}_\phi[a,\tilde{\phi},\{\tilde{\varphi}_m\},\tilde{R}] \equiv a^{1+\alpha}\tilde{\nabla}^2\tilde{\phi} - a^{3+\alpha}\left(\xi\tilde{R}\tilde{\phi} + \frac{\partial\tilde{V}}{\partial\tilde{\phi}}\right). \end{cases} \tag{26}$$

Similarly, to evolve the scale factor we use Eq. (21) as derived from the trace of the energy-momentum tensor. Defining the conjugate momentum of $a(\eta)$ as

$$\pi_a = a^{1-\alpha}a'\,, \tag{27}$$

we arrive to a system of coupled first-order differential equations depending on another kernel functional,

$$\begin{cases} a' = a^{\alpha-1}\tilde{\pi}_a\,, \\ \tilde{\pi}'_a = \tilde{\mathcal{K}}_a[a,\tilde{R}], \quad \text{with} \quad \tilde{\mathcal{K}}_a[a,\tilde{R}] \equiv \frac{a^{2+\alpha}}{6}\tilde{R}. \end{cases} \tag{28}$$

To close the system, an expression for $\tilde{R}$ is needed in both kernels $\mathcal{K}_\phi$, $\mathcal{K}_a$. Using Eq. (20), we can write

$$\tilde{R} = \frac{f_*^2}{m_p^2}\left[\frac{2(1-6\xi)\left(\tilde{E}_G^{\tilde{\phi}} - \tilde{E}_K^{\tilde{\phi}}\right) + 4\langle\tilde{V}\rangle - 6\xi\langle\tilde{\phi}\,\tilde{V}_{,\tilde{\phi}}\rangle + (\tilde{\rho}_m - 3\tilde{p}_m)}{1 + (6\xi-1)\xi\langle\tilde{\phi}^2\rangle f_*^2/m_p^2}\right], \tag{29}$$

where we have used $\langle T_m\rangle = 3p_m - \rho_m$ and introduced the volume-averaged kinetic $\tilde{E}_K^{\tilde{\phi}}$ and gradient $\tilde{E}_G^{\tilde{\phi}}$ energy densities

$$\tilde{E}_K^{\tilde{\phi}} = \frac{1}{2a^6}\langle\tilde{\pi}_{\tilde{\phi}}^2\rangle\,, \qquad \tilde{E}_G^{\tilde{\phi}} = \frac{1}{2a^2}\sum_i\langle\tilde{\partial}_i\tilde{\phi}\,\partial_i\tilde{\phi}\rangle\,. \tag{30}$$

In summary, Eqs. (26) and (28), together with the expression for $\tilde{R}$ in Eq. (29) (plus the equations of motion of the unspecified matter sector), represent a set of equations that completely characterizes the dynamics of a system with a scalar field non-minimally coupled to gravity in the Jordan frame. Generalization to multiple non-minimally coupled scalars is obtained straight forwardly by summing over the terms with non-minimal coupling $\xi_i\phi_i^2$ in Eq. (29).

## 3 Lattice formulation

In order to evolve our system of equations Eqs. (26), (28) and (29) in a way that fully captures the spatial dependence of the fields, we need to choose a time evolution scheme and to introduce a spatial discretization prescription. We use a lattice with $N$ sites per dimension with periodic boundary conditions. We will consider the lattice sites to represent comoving coordinates. If the (comoving) length of the grid is $L$, the resulting (comoving) lattice spacing

between sites is $\delta x = L/N$. We work with finite differences and use the following notation for the forward and backward derivatives

$$\nabla_i^{\pm} f(\mathbf{n}) = \frac{\pm f(\mathbf{n}) \mp f(\mathbf{n} \pm \hat{i})}{\delta x}, \tag{31}$$

where $f$ is an arbitrary scalar function defined on the lattice sites $\mathbf{n} = (n_1, n_2, n_3)$, and $\hat{i}$ represents a displacement vector of one unit in the $i$-th direction. We discretize the gradient terms using forward differences and the Laplacian using a symmetric discretization

$$\sum_i \langle \partial_i \phi \partial_i \phi \rangle \longrightarrow \sum_i \langle \nabla_i^+ \phi \nabla_i^+ \phi \rangle, \tag{32}$$

$$\vec{\nabla}^2 \phi \longrightarrow \sum_i \nabla_i^- \nabla_i^+ \phi. \tag{33}$$

We are now in a position to define the evolution equations by introducing the discrete kernels

$$\tilde{\mathcal{K}}_{\phi}\left[a, \tilde{\phi}, \{\tilde{\varphi}_{\mathrm{m}}\}, \tilde{R}\right] = a^{1+\alpha} \tilde{\nabla}_i^- \tilde{\nabla}_i^+ \tilde{\phi} - a^{3+\alpha}\left(\xi \tilde{R} \tilde{\phi} + \frac{\partial \tilde{V}}{\partial \tilde{\phi}}\right), \tag{34}$$

$$\tilde{\mathcal{K}}_a\left[a, \tilde{R}\right] = \frac{a^{2+\alpha}}{6} \tilde{R}, \tag{35}$$

which we have already written in terms of natural field and spacetime variables, c.f. Eq. (22). We have also introduced dimensionless discrete derivatives $\tilde{\nabla}$ given by Eq. (31) in terms of the dimensionless lattice spacing $\delta \tilde{x} = \tilde{L}/N = \omega_* \delta x$, with $\tilde{L} = \omega_* L$.

At this point, it is important to realize that $\tilde{R} = \tilde{R}[\tilde{\phi}, \tilde{\pi}_{\phi}, \{\tilde{\varphi}_{\mathrm{m}}\}, \{\tilde{\pi}_{\varphi_{\mathrm{m}}}\}]$ depends on all fields and conjugate momentum variables, and hence the kernel for the non-minimally coupled field $\tilde{\phi}$ depends on its own conjugate momentum. Because of this, preferred symplectic algorithms such as *staggered Leapfrog*, *velocity-* or *position-Verlet*, cannot be used (see Ref. [67] for a discussion on this). We can instead use *Runge-Kutta* (RK) methods, in particular explicit RK algorithms. We have adapted the well known *mid-point method* to our set of equations, corresponding to a second order RK method. To account for situations where a high time-accuracy may be required, we have also implemented a particularly interesting family of explicit *low-storage* RK methods of higher order following Refs. [68, 69]. These present multiple advantages: they are easy to implement, the memory cost does not increase when increasing the accuracy order, and in some cases an adaptive time-step scheme is allowed. The interested reader can find more information and an explicit description of all these RK algorithms applied to our system of equations in Appendix C.

One last important point is to have a discrete version of the Hubble constraint given in Eq. (13). Verifying that this constraint is preserved by our numerical evolution scheme provides an important check of the method (the resulting convergence is shown in Appendix D). In terms of rescaled variables, it reads

$$\left(\frac{a'}{a}\right)^2 = \frac{a^{2\alpha} f_*^2}{3m_p^2}\left[\tilde{\rho}_{\mathrm{m}} + \tilde{E}_K^{\tilde{\phi}} + \tilde{E}_G^{\tilde{\phi}} + \langle \tilde{V} \rangle + \frac{3\xi}{a^{2\alpha}}\left(\frac{a'}{a}\right)^2 \langle \tilde{\phi}^2 \rangle + \frac{6\xi}{a^{\alpha+3}}\left(\frac{a'}{a}\right) \langle \tilde{\phi} \tilde{\pi}_{\tilde{\phi}} \rangle\right], \tag{36}$$

where we have dropped the $\langle \nabla^2 \phi^2 \rangle$ term of Eq. (17) because it is a total derivative whose volume average vanishes due to the periodic boundary conditions of the lattice. We now have all the tools to evolve our system of equations on the lattice. In the next section, we present an explicit example in the context of geometric preheating. Lastly, note that all numerical algorithms presented above have been implemented in the package $\mathcal{C}$osmo$\mathcal{L}$attice [67, 70], which can perform user-friendly and versatile field theory simulations. These new algorithms will be made publicly available in a future update of $\mathcal{C}$osmo$\mathcal{L}$attice.

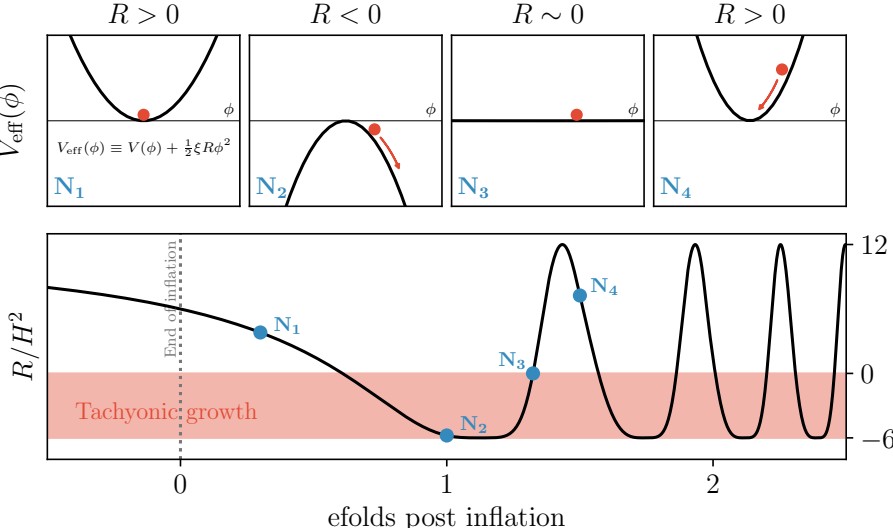

Figure 1: Time evolution of the effective potential of the non-minimally coupled field $\phi$ as a function of the Ricci scalar. Oscillations between an unbounded and bounded potential occur once the Ricci scalar begins oscillating at around one e-fold.

# 4  Example: Geometric preheating

We now study an example of geometric preheating directly in the Jordan frame, using the formalism developed in the previous sections. By geometric preheating, we refer to the excitation of a light spectator field $\phi$ non-minimally coupled to gravity. This occurs due to the oscillatory behavior of the spacetime curvature $R$ that follows after inflation, when a homogeneous inflaton field oscillates around the minimum of its potential [28] illustrated in Fig. 1. The fact that $R$ becomes oscillatory can be seen from the traced Einstein equations, assuming that the homogeneous inflaton field $\chi$ initially dominates the energy density of the universe, such that $T = T_\phi + T_{\text{inf}} \approx T_{\text{inf}} = -\partial^\mu \chi \partial_\mu \chi - 4V_{\text{inf}}(\chi)$. This leads to

$$R = -\frac{T}{m_p^2} \approx \frac{1}{m_p^2}\left[\partial^\mu \chi \partial_\mu \chi + 4V_{\text{inf}}(\chi)\right] = \frac{1}{m_p^2}\left[4V_{\text{inf}}(\chi) - \frac{1}{a^{2\alpha}}\chi'^2\right]. \tag{37}$$

One illustrative example is the case where $V_{\text{inf}}(\chi) = \frac{1}{2}m^2\chi^2$, in which case $R$ can be approximated in cosmic time ($\alpha = 0$) as

$$R = \frac{1}{m_p^2}\left[4V_{\text{inf}}(\chi) - \dot\chi^2\right] \approx \frac{R_0}{4}\left(\frac{a_0}{a}\right)^3\left[1 + 3\cos(2mt)\right]. \tag{38}$$

In this expression, it is manifest that $R$ oscillates between positive and negative values due to the harmonic oscillations of $\chi$. In general, for an inflaton potential with a minimum around the origin and an arbitrary power law behavior $V_{\text{inf}}(\chi) \propto |\chi|^p$ ($p > 1$), (or even for a linear combination of various power laws) the oscillations of the inflaton will not be harmonic. This does not change the fact that $R$, and hence the effective mass squared of the spectator field $m_{\phi,\text{eff}}^2 = \xi R$, will still alternate periodically between positive and negative values. As a consequence of the periodic tachyonic stages ($m_{\phi,\text{eff}}^2 < 0$), initial quantum vacuum fluctuations of the spectator field $\phi$ can be exponentially amplified if the strength of its non-minimal coupling $\xi$ is large enough. The amplification may persist until the effective tachyonic mass of $\phi$ is fully screened by its own self-interactions, or until the energy of $\phi$ grows to the same order as the energy available in the system. In either case, a detailed lattice study is required due to the

non-linearity of the system. We present first in Section 4.1 a linear analysis of the initial insta-
bility of the mode functions of $\phi$, then in Section 4.2 we present an analysis of the evolution
of the system once the dynamics become non-linear.

## 4.1 Initial conditions via linear analysis

Our procedure will consist of computing the power spectrum of the $\phi$ fluctuations induced
during inflation and the subsequent transition period via a linear analysis, which we then
use as the initial condition for the lattice evolution before the dynamics enter the non-linear
regime. To proceed, we consider a theory involving an inflaton field $\chi$ and a light spectator
field $\phi$ with a non-minimal coupling to gravity, similar to Refs. [28, 30, 31, 71]

$$S = \int d^4x \sqrt{-g} \left[ \frac{m_p^2}{2} R + \mathcal{L}_\phi + \mathcal{L}_{\rm inf} \right], \tag{39}$$

with

$$\mathcal{L}_\phi = -\frac{1}{2} g^{\mu\nu} \partial_\mu \phi \partial_\nu \phi - \frac{1}{2} \xi R \phi^2 - V(\phi), \tag{40}$$

$$\mathcal{L}_{\rm inf} = -\frac{1}{2} g^{\mu\nu} \partial_\mu \chi \partial_\nu \chi - V_{\rm inf}(\chi). \tag{41}$$

In this theory, the inflaton $\chi$ and the spectator field $\phi$ interact only gravitationally through
the non-minimal coupling $\xi R \phi^2$. During slow-roll inflation, we have a quasi de-Sitter phase
where $R \approx 12H^2$ and $H \approx$ constant (its time derivative is slow-roll suppressed). This means
that for $\xi > 0$, the spectator field has a heavy effective mass $m_{\phi,\rm eff}^2 \approx 12\xi H^2$ during inflation.
We assume that this effective mass dominates over $V''(\phi)$, such that the potential $V(\phi)$ can be
neglected during inflation. This, combined with the fact that the non-minimally coupled spec-
tator field $\phi$ is energetically subdominant during inflation, justifies the use of a linear analysis.
It will be convenient to work in conformal time ($\alpha = 1$) where the metric is conformally flat
and quantization proceeds as in Minkowski space. In that case, we can write the action for $\phi$
in terms of the canonically normalized field $\varphi = a\phi$

$$S_\varphi = \frac{1}{2} \int d\tau d^3x \left[ (\varphi')^2 - (\nabla\varphi)^2 - a^2 \left( \xi - \frac{1}{6} \right) R \varphi^2 \right]. \tag{42}$$

We then canonically quantize $\varphi$ as

$$\hat{\varphi}(x,\tau) = \int \frac{d^3k}{(2\pi)^3} \left[ \varphi_k(\tau) \hat{a}_{\mathbf{k}} e^{i\mathbf{k}\cdot\mathbf{x}} + \varphi_k^*(\tau) \hat{a}_{\mathbf{k}}^\dagger e^{-i\mathbf{k}\cdot\mathbf{x}} \right], \tag{43}$$

where $[\hat{a}_{\mathbf{k}}, \hat{a}_{\mathbf{k}'}^\dagger] = (2\pi)^3 \delta(\mathbf{k} - \mathbf{k}')$ and the modes are normalized such that $\varphi_k \varphi_k'^* - \varphi_k' \varphi_k^* = i$.
The mode functions $\varphi_k(\tau)$ obey the equation of motion given by Eq. (42) in momentum space

$$\varphi_k''(\tau) + \left[ k^2 + a^2 \left( \xi - \frac{1}{6} \right) R \right] \varphi_k(\tau) = 0. \tag{44}$$

We assume that the evolution of each mode starts far inside the Hubble radius, namely
$-k\tau \gg 1$, where the curvature is negligible. In that case, the modes should approach the
*Bunch-Davies* vacuum

$$\varphi_k(k\tau \to -\infty) = \frac{1}{\sqrt{2k}} e^{-ik\tau}. \tag{45}$$

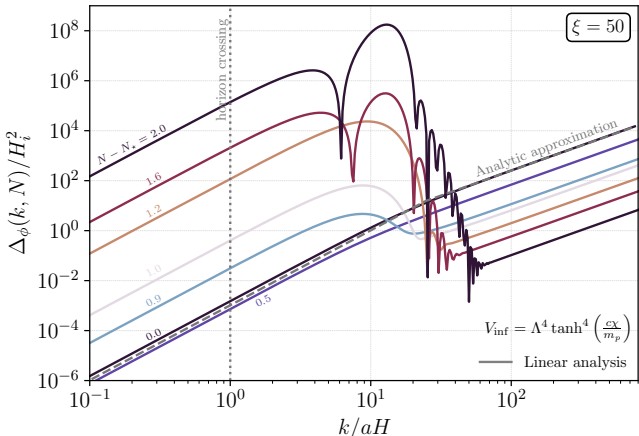

Figure 2: Resulting power spectrum of the non-minimally coupled spectator field from the linear analysis performed in Section 4.1. Colored lines illustrate the spectrum at the indicated number of e-folds post inflation.

We are interested in the power spectrum $\Delta_\varphi(k,\tau)$, which in terms of the two-point function is defined as

$$\langle \varphi^2 \rangle = \langle 0|\hat{\varphi}(\tau,x)\hat{\varphi}(\tau,x)|0\rangle = \int \frac{dk}{k} \Delta_\varphi(k,\tau). \tag{46}$$

The power spectrum of the original field $\phi$ is then related as

$$\Delta_\phi(k,\tau) = \frac{1}{a^2}\Delta_\varphi(k,\tau) = \frac{k^3}{2\pi^2 a^2}|\varphi_k(\tau)|^2 \equiv \frac{k^3}{2\pi^2}\mathcal{P}_\phi(k,\tau). \tag{47}$$

For our numerical results (which involve integrating over many e-folds of inflaton), it proves easier to solve Eq. (44) in cosmic time ($\alpha = 0$), where it reads

$$\ddot{\varphi}_k + H\dot{\varphi}_k + \left[\frac{k^2}{a^2} + \left(\xi - \frac{1}{6}\right)R\right]\varphi_k = 0, \tag{48}$$

with the Bunch-Davies initial condition now expressed as

$$\varphi_k(k/(aH) \gg 1) \approx \frac{1}{\sqrt{2k}}e^{\frac{ik}{aH}}, \tag{49}$$

where we have used $\tau \approx -1/(aH)$, given that $H$ changes very slowly. As previously mentioned, the non-minimally coupled spectator field $\phi$ is initially energetically subdominant by assumption, so we neglect its contribution to the background evolution for the linear analysis. In this case, all modes evolve independently and the background evolution is completely determined by the homogeneous energy components of the inflaton

$$H^2 \equiv \left(\frac{\dot{a}}{a}\right)^2 = \frac{1}{3m_p^2}\left(\frac{1}{2}\dot{\chi}^2 + V_{\text{inf}}(\chi)\right), \tag{50}$$

with the evolution of the homogenous inflaton governed by the standard Klein-Gordon equation of motion in cosmic time

$$\ddot{\chi} + 3H\dot{\chi} + \frac{\partial V_{\text{inf}}}{\partial \chi} = 0. \tag{51}$$

In our numerical analysis we consider an observationally viable inflationary model inspired by *α-attractors* [72], with inflaton potential parametrised as [21]

$$V_{\text{inf}}(\chi) = \Lambda^4 \tanh^p\left(\frac{c\chi}{m_p}\right), \quad \text{with} \quad p = 4, 6, \tag{52}$$

which flattens out for $|\chi| \gg m_p/c$ (where $c$ is a dimensionless parameter), and takes a power-law form $V \propto \chi^p$ for $|\chi| \ll m_p/c$. We take $c = 0.1$ which reproduces the observed value of the scalar perturbations at CMB scales for $V_{\text{inf}}(\chi_{\text{CMB}}) = (1.6 \times 10^{16}\,\text{GeV})^4$ and saturates the upper bound on the scale of inflation [21] corresponding to $\Lambda = 1.79 \times 10^{16}$ GeV.

We solve Eq. (48) numerically by discretizing $k$ on a grid of 512 log-spaced modes. We begin evolving each mode considering the Bunch-Davies initial condition when $k/(aH) = \beta$, with $\beta \gg 1$ a penetration factor. Larger values of $\beta$ better approximate the Bunch-Davies initial condition, but also increase simulation time, so as a compromise we choose $\beta = 10^3$. At the end of inflation, we would like to have a superhorizon power spectrum of simulated modes spanning at least three orders of magnitude in $k$-space. Since the lowest $k$ mode starts a factor $\beta$ inside the horizon, we require approximately $\Delta N \simeq \log(10^3\beta) \approx 14$ e-folds of simulated inflation for all modes of interest to exit the horizon. We therefore choose the initial conditions of the homogeneous inflaton field $\chi_i$ such that we obtain 14 e-folds of inflation (this corresponds to $\chi_i = 9.23(10.87)m_p$ for $p = 4(6)$, respectively). Following this procedure, we numerically integrate Eqs. (48), (50) and (51) for 14 e-folds of inflation and through the transition to the post-inflationary stage. We show the resulting power spectrum in Fig. 2 for $\xi = 50$.

For comparison, our numerical results for $\phi$ can be compared to the predicted power spectrum during inflation in pure de-Sitter space, which was computed analytically in Ref. [61] as

$$H_*^{-2}\Delta_\phi(z) = \frac{z^3}{8\pi}|H_{i\mu}^{(1)}(z)|^2 e^{-\pi\mu} \approx \frac{1}{4\pi^2}\begin{cases}\frac{z^3}{\mu}, & z \ll 1 \quad \text{(superhorizon)}, \\ z^2, & z \gg 1 \quad \text{(subhorizon)}.\end{cases} \tag{53}$$

Here $z = k/(aH_*)$, $\mu^2 = 12(\xi - 3/16)$, and $H_*$ is taken to be the Hubble rate at the end of inflation. The approximate equality holds for $\xi > 3/16$. According to this expression, we see that the superhorizon fluctuations during inflation follow a $k^3$ power law. This is expected as after Hubble radius exit the modes are damped because of the heavy effective mass induced by the non-minimal coupling. On the other hand, the subhorizon modes deep inside the Hubble radius remain in the Bunch-Davies vacuum $\propto k^2$, indicating that they are not excited. The transition between power laws occurs around $k/(aH_*) \approx \mu$, where we have $\mu \approx \sqrt{12\xi}$ for $\xi \gg 1$. We see that this analytic approximation explains well the behavior of the power spectrum shown in Fig. 2 at the end of inflation.

After inflation ends, the inflaton begins oscillating around the minimum of its potential, which also induces oscillations in $R$, as shown in Fig. 1. This allows for tachyonic growth of the non-minimally coupled field during the periods when $R < 0$, leading to the emergence of a peak in its power spectrum after inflation. That fact that the peak is stationary when the power spectrum is plotted in terms of $k/(aH)$ can be obtained from inspecting Eq. (48), as the tachyon is regularized (on a per mode basis) when $k/(aH) \approx \sqrt{|\xi R/H^2|} \approx \mathcal{O}(1)\sqrt{\xi} \lesssim \sqrt{6\xi}$, where we used that tachyonic growth happens for $-6 \leq R/H^2 < 0$ and assumed $\xi \gg 1$. Correspondingly, the peak in the power spectrum, which clearly emerges about $\sim 1$ e-fold after the end of inflation, is observed at $k/(aH) \approx \mathcal{O}(1)\sqrt{\xi}$. It is at this moment when we introduce the power spectrum from our linear analysis, in order to initialize the non-minimally coupled field in the lattice simulation. The shape of the peak of the power spectrum determines the range of comoving momenta we need to consider on the lattice. In terms of comoving momenta, the peak shifts to smaller values of $k$ while its amplitude grows during the linear

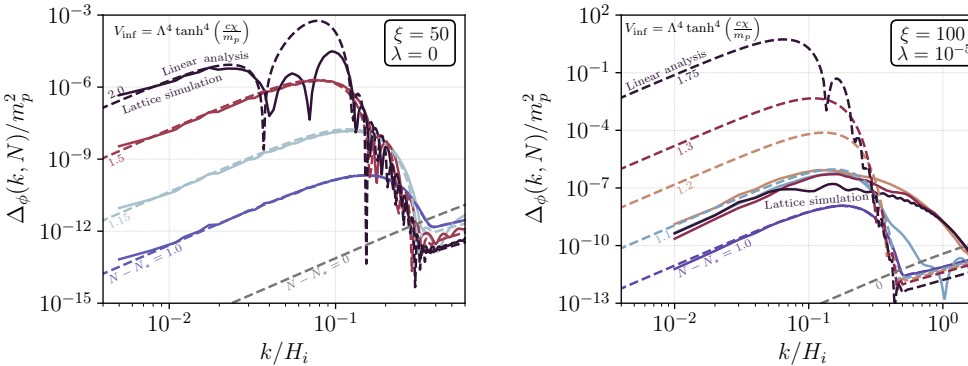

Figure 3: Power spectra of the non-minimally coupled spectator field $\phi$ at the end of inflation defined by $N = N_*$ and $N - N_*$ e-folds thereafter. The solid/dashed lines are the results of the lattice/linear simulations. **Left:** Power spectrum for the case of $\xi = 50$ where the spectator field has no potential, $V = 0$. Initially the spectrum is well described by the linear analysis until the growth becomes large enough that backreaction occurs at approximately $N - N_* = 1.75$. **Right:** Power spectrum for $\xi = 100$ where the spectator field has a small quartic coupling $\lambda = 10^{-5}$. Deviations from the linear analysis begin almost immediately where we see the non-linear effect of the quartic term transferring power to higher $k$-modes.

regime. Hence, the most important scales to capture in the lattice are those spanned by the peak itself and its infrared tail to some extent, so there is room for the peak to shift further to the infrared as we simulate the dynamics in the lattice.

## 4.2 Non-linear lattice analysis

After a clear peak in the power spectrum emerges in the linear analysis, but still before any interactions of $\phi$ or its backreaction onto the background dynamics are relevant, we move to solving the system on the lattice. In particular, we treat $\phi$ as a classical field whose initial fluctuations are drawn randomly from a Gaussian distribution with power spectrum given by $\Delta_\phi$ that we computed in the linear analysis up to one e-fold after the end of inflation. We now fully include all interactions as well as a self-consistent evolution of the expanding background including contributions by all fields present. The relevant equations of motions to solve are

$$\phi'' + (3-\alpha)\frac{a'}{a}\phi' - \frac{\nabla^2\phi}{a^{2(1-\alpha)}} + a^{2\alpha}\xi R\phi = -a^{2\alpha}\frac{\partial V}{\partial \phi}, \tag{54}$$

$$\chi'' + (3-\alpha)\frac{a'}{a}\chi' - \frac{\nabla^2\chi}{a^{2(1-\alpha)}} = -a^{2\alpha}\frac{\partial V_{\text{inf}}}{\partial \chi}, \tag{55}$$

$$\frac{a''}{a} + (1-\alpha)\left(\frac{a'}{a}\right)^2 = \frac{a^{2\alpha}}{6}R, \tag{56}$$

where Eq. (56) determines the background evolution as discussed in Section 2 and $R$ is defined as

$$R = \frac{F(\phi)}{m_p^2}\Big[(6\xi - 1)\left(\frac{1}{a^{2\alpha}}\langle\phi'^2\rangle - \frac{1}{a^2}\langle(\nabla\phi)^2\rangle\right)$$

$$- 6\xi\langle\phi V_{,\phi}\rangle + 4\langle V + V_{\text{inf}}\rangle - \frac{1}{a^{2\alpha}}\langle\chi'^2\rangle + \frac{1}{a^2}\langle(\nabla\chi)^2\rangle\Big], \tag{57}$$

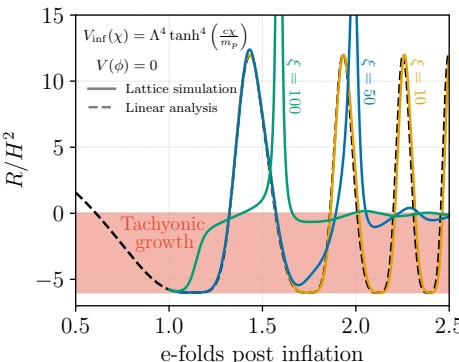
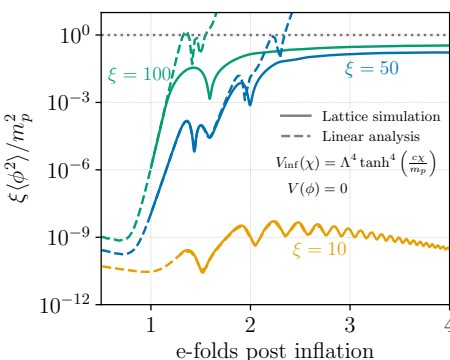

Figure 4: **Left:** Evolution of the Ricci scalar normalized to Hubble squared $H(t)^2$. The dashed-black line is the evolution of the linear free-field analysis which is used as an input to the lattice simulation. The colored lines $\xi = \{10, 50, 100\}$, {yellow, blue, green} are the lattice simulation results which illustrate a strong deviation once the energy density of the non-minimally coupled field is comparable to that of the inflaton. Note that the two peaks extend to values outside the range of the figure, $R/H^2 = 20.8\,(49.1)$ for $\xi = 50\,(100)$, respectively. Shaded in red is the region where the Ricci scalar is negative, driving tachyonic growth of the spectator field. **Right:** Expectation value $\langle \phi^2 \rangle$, see Eq. (46), of the non-minimally coupled spectator field for the same values of $\xi$ as before.

with $F(\phi)$ given in Eq. (20). We adapt the continuum equations to the lattice following Section 3 and Appendix C. All simulations with $\lambda = 0$ are run on lattices of size $N = 240$ points per spatial dimension and evolved with the RK2MP method described in detail in Appendix C. In the lattice, we use natural variables as defined in Eq. (22), with $f_* = m_p$ and $\omega_* = H_i$, where $H_i$ is the Hubble rate at the start of the linear analysis $\simeq 14$ e-folds before the end of inflation. Note that we are using time-steps of $H_i \delta t = 0.01$ in the evolution with $k_{\rm IR}/H_i = 2.5 \times 10^{-3}\,(4 \times 10^{-3})$ for $\xi = 10\,(50 \text{ or } 100)$. In the case $\lambda = 1 \times 10^{-5}$, we used lattices of size $N = 512$ and $k_{\rm IR}/H_i = 1 \times 10^{-2}$.

Let us first consider the case where $V(\phi) = 0$. The resulting evolution of the Ricci scalar in the $p = 4$ case for the inflaton potential is illustrated by the dashed-black line in the left-hand panel of Fig. 4. Here we see that shortly after the end of inflation the value of $R/H^2$ tends to negative values before oscillating in the range $-6 \leq R/H^2 \leq 12$. While $R$ is negative, the non-minimally coupled field has a tachyonic effective mass and can experience exponential growth. The structure of this growth can be seen in the power spectrum of the left-hand panel of Fig. 3, where different line colors indicate the number of efolds post inflation. Here we see the growth of the peak agrees well between the linear (dashed) and lattice (solid) results so long that the energy density in the NMC field is sub-dominant. Departure from the linear analysis can be more easily seen in the right-hand panel of Fig. 4 where we see the expectation value $\langle \phi^2 \rangle$ growing exponentially when $R$ takes negative values, as expected. For comparison purposes, the dashed lines show the results of the linear analysis, where the field grows unbounded when the tachyonic growth is strong enough to overcome the expansion of the universe (as in the $\xi = 50, 100$ cases) since there we neglected the contribution of $\phi$ to the background evolution. This is not the case in the lattice analysis, where for these large values of $\xi$ the backreaction of $\phi$ on the background evolution asymptotically drives $\xi \langle \phi^2 \rangle / m_p^2$ to a constant value below unity, and $R$ ends oscillating around zero with a damped amplitude, as shown by the solid lines in Fig. 4. More specifically, at the onset of backreaction the kinetic term $(6\xi - 1)\langle \phi'^2 \rangle$ drives initially $R$ to a large positive value, as represented by the green (blue)

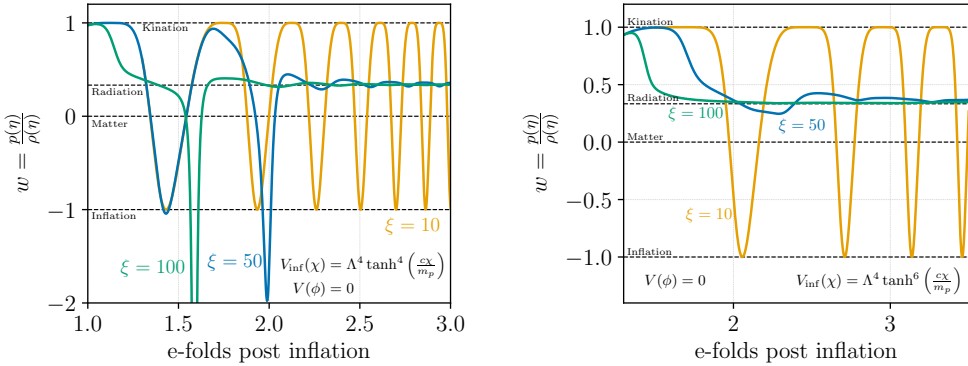

Figure 5: Evolution of the total equation of state as a function of e-folds post inflation. Here we define $w \equiv p(\eta)/\rho(\eta)$ using the non-minimally coupled fields contributions from Eq. (17) and Eq. (18) plus the standard minimally coupled contributions from the inflaton. We show the results for $p = 4$ ($p = 6$) on the left (right), respectively. In the right-hand panel the $w$ peaks extends to $w = -2(-5.1)$ for $\xi = 50(100)$.

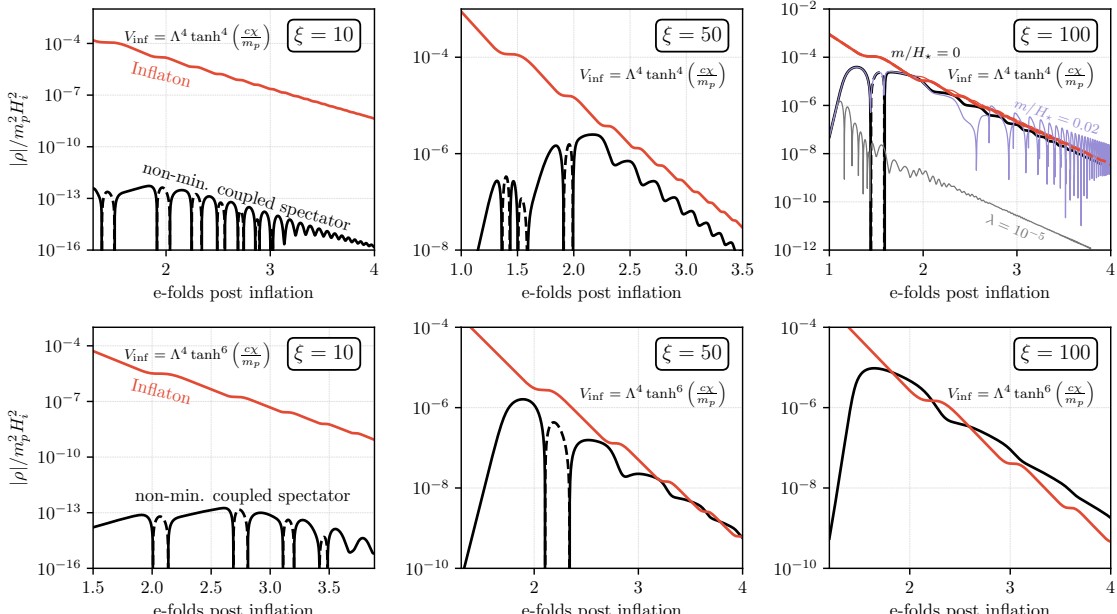

Figure 6: Time evolution of the energy density of both the inflaton (red) and non-minimally coupled spectator field (black). Parameter values for the inflationary potential are $c = 0.1$, $\Lambda \simeq 1.79 \times 10^{16}$ GeV, while the energy density is normalized as in Eq. (24). We show the absolute value of the energy density but change the line style to dashed to illustrate when the energy density turns negative. **Top row:** Three values of the non-minimal coupling $\xi$ for the inflationary potential with $p = 4$. In the top-right panel we also show the effect of a Hubble scale mass (thin purple) and a small quartic coupling of the spectator field (thin gray). **Bottom row:** Same three values of $\xi$ but for $p = 6$, where the energy density of the inflaton redshifts faster than radiation.

spikes for $\xi = 100$ (50) in the left panel of Fig. 4. This results in a large positive effective mass squared $\xi R$, that induces a restoring force for $\phi$, opposing its growth. The field velocity $\phi'$ is then suppressed causing $R$ to start a rapid descent down to a small negative value, after which it begins oscillating around zero. After the spike, $R$ oscillates with a small damped amplitude, so the successive tachyonic mass stages cannot overtake anymore the expansion of the universe, and $\xi\langle\phi^2\rangle/m_p^2$ approaches asymptotically a constant value. Returning to the power spectrum (left-hand panel of Fig. 3) at $N = 2$ differences in both the peak and also the UV tail of the spectrum arise. The origin of the additional structure in the peak of the lattice results is simply resultant from the Ricci scalar remaining positive once the backreaction occurs. Subsequently the NMC just behaves as a free oscillator and is no longer driven. If we define the equation of state in terms of the total pressure and energy densities $w = p(\eta)/\rho(\eta)$, then $w$ can be written in terms of $R$ by combining Eqs. (2), (13) and (14)

$$w = \frac{p(\eta)}{\rho(\eta)} = \frac{1}{3}\left(1 - \frac{R}{3H^2}\right), \tag{58}$$

where $H = \mathcal{H}/a^\alpha$. We see that the large positive spikes where $R/H^2 > 12$ when the non-minimally coupled field backreacts correspond to periods where the equation of state spikes below $w = -1$, as shown in the left-hand figure of Fig. 5. Since $\rho$ is always constrained to be positive definite by Eq. (13), it can be seen from Eq. (58) that $R/H^2 > 12$ corresponds to a large negative pressure, namely $p < -\rho$, which violates the classical energy conditions due to the non-minimal interaction of $\phi$ with the gravitational field. In particular, the dominant contribution to the pressure during these spikes comes from the $(1-4\xi)\langle\phi'^2\rangle$ term in Eq. (18), which is always negative for $\xi > 1/4$.

Before turning to the evolution of the energy density, a comment on the effect of a quartic interaction term in the scalar potential is in order. In the right-hand panel of Fig. 3 we show the power spectrum for the case where $\xi = 100$ with a quartic $\lambda = 10^{-5}$. As expected re-scattering of modes leads to additional power in the UV spectrum while also screening the effect of the tachyon, namely the growth of the peak is diminished already at $N = 1.1$.

Finally, in Fig. 6 we show the evolution of the energy density of both the inflaton $\chi$ (red) and the non-minimally coupled spectator field $\phi$ (black). In the top (bottom) row we consider the hypertangent inflaton potential with $p = 4$ ($p = 6$), while in the three columns we again consider the three benchmark values of $\xi = \{10, 50, 100\}$. In the $p = 4$ case, the inflaton energy density drops like radiation since the potential is quartic around the minimum, while in the $p = 6$ case the inflaton energy density decays faster than radiation. Though the total energy density is always positive, it is well known that the energy density of the non-minimally coupled field as defined in Eq. (17) is not positive definite and we indicate when it becomes negative using dashed lines. In the cases where $V(\phi) = 0$, the energy density of the non-minimally coupled field scales as radiation at late times, as can be seen in Fig. 5. In the upper right-hand panel of Fig. 6 we also show the behavior when the non-minimally coupled field has a non-zero potential $V(\phi)$. We consider the cases of $V(\phi) = m^2\phi^2/2$ with mass $m/H_i = 0.02$ (thin purple line) as well as $V(\phi) = \lambda\phi^4/4$ with a quartic $\lambda = 10^{-5}$ (thin gray line). The case $V(\phi) = m^2\phi^2/2$ exhibits similar behavior to the $V(\phi) = 0$ case until around 2 e-folds where bare mass term begins to dominate over the effective mass induced by $R$ and the energy density of the non-minimally coupled field begins to scale like matter. This allows the energy density of the non-minimally coupled field to dominate over that of the inflaton, providing an efficient reheating mechanism. The effect of the quartic is much more drastic because it acts to regulate the tachyonic growth occurring when $R$ is negative, preventing the non-minimally coupled field from reaching large field values. After a transition period where the energy density dilutes faster than radiation due to the $\xi$ dependent terms in Eq. (17), the energy becomes dominated by the field oscillating in its quartic potential which leads to

radiation scaling of the energy density. This case does not result in the non-minimally coupled field fully reheating the universe unless the inflaton energy drops faster than radiation. This is precisely what occurs in the $p = 6$ case, shown in the bottom row of Fig. 6. In this case, the energy density of the non-minimally coupled field can quickly come to dominate over that of the inflaton.

## 5 Summary and conclusions

The presence of at least one fundamental scalar field in the SM raises the question of the role non-minimal couplings to gravity may play in the evolution of the early Universe. Any scalar $\phi$ in curved spacetime, be it the SM Higgs or otherwise, inevitably acquires a non-minimal coupling to gravity of the form $\xi|\phi|^2 R$ through renormalization group evolution. Typically, the dynamics of non-minimally coupled scalars are not studied directly in the original Jordan frame, but rather in the Einstein frame via a conformal transformation of the metric that brings the action to the canonical Einstein-Hilbert form. This approach allows for a more intuitive interpretation of the dynamics, however, the equivalence of these two frames in situations where the initial conditions are set by quantum fluctuations is unclear.

In contrast, in this work we have developed an approach to solve the dynamics of non-minimally coupled scalars in an expanding universe directly as written in the original Jordan frame, where the non-minimal couplings are maintained explicitly. In the Jordan frame, the equations of motion describing the background evolution are typically non-linear in the derivatives of the scale factor, making them difficult to solve in practice. We tackle this problem by considering the trace of the energy-momentum tensor, a simpler object that can be related to the background evolution. This admits a simple system of coupled first-order differential equations for the background evolution that can be straightforwardly numerically integrated. In Section 3, we demonstrate how this method can be implemented in the $\mathcal{C}$osmo$\mathcal{L}$attice [67, 70] package by specifying the discrete evolution kernels. There, we see that the resulting kernel for the non-minimally coupled field evolution depends on its own conjugate momentum, preventing the use of symplectic algorithms typically employed. We have therefore implemented explicit "low-storage" Runge-Kutta methods that allow for high-order methods while keeping memory usage constant and permitting adaptive time-steps, see appendix C for further details.

To demonstrate the viability of our method, we study geometric preheating as an illustrative example in Section 4. The model involves a real spectator scalar field $\phi$ that is excited through its non-minimal coupling to gravity when the inflaton oscillates around the minimum of its potential following the end of inflation. The oscillations of the inflaton source oscillations in $R$, inducing a time-dependent effective mass for $\phi$. While the effective squared mass is negative, tachyonic growth of the non-minimally coupled field can occur if the value of $\xi$ is large enough to overcome the friction due to the expansion of the universe, as shown in Fig. 6. In this case, the growth of the non-minimally coupled field is highly efficient and the energy density of $\phi$ reaches an $\mathcal{O}(1)$ value of the total energy density within an $\mathcal{O}(1)$ number of e-folds, representing an extremely efficient preheating mechanism. We find that if there is no explicit scale in the potential of the non-minimally coupled field, then its energy density scales as radiation at late times. Therefore, in the cases where $\xi$ is large enough, whether the inflaton or non-minimally coupled field dominates the energy density at late times depends on the choice of potential for both fields.

To conclude, we have introduced a robust method to solve the dynamics of non-minimally coupled scalar fields directly in the original Jordan frame with the expansion sourced by all fields present, even when the dynamics becomes fully inhomogeneous and/or non-linear due to the backreaction of the excited species. This will provide an important tool to study the

equivalence of the Einstein and Jordan frame, in particular when the initial conditions of the fields are set by quantum vacuum fluctuations in both frames. All numerical algorithms presented in this paper will be made publicly available in a future update to the package CosmoLattice, which will allow anyone to perform user-friendly and versatile cosmological simulations involving non-minimally coupled scalar fields.

## Acknowledgments

We would like to thank Nicolás Loayza for useful numerical assistance and for validating a number of results presented here. In addition, TO would also like to thank Valerie Domcke, Alan Guth, David I. Kaiser, Nadav Outmezguine and Marko Simonovic for insightful discussions.

**Funding information** DGF (ORCID 0000-0002-4005-8915) is supported by a Ramón y Cajal contract with Ref. RYC-2017-23493. This work is also supported by project PROMETEO/2021/083 from Generalitat Valenciana, and by project PID2020-113644GB-I00 from Ministerio de Ciencia e Innovación.

## A Curvature in $\alpha$-time

Given a flat "$\alpha$-time" FLRW background described by the metric

$$ds^2 = -a(\eta)^{2\alpha}d\eta^2 + a(\eta)^2 \delta_{ij} dx^i dx^j\,, \tag{A.1}$$

the Christoffel symbols can be computed from

$$\Gamma^{\sigma}_{\mu\nu} = \frac{1}{2} g^{\sigma\rho} \left( \partial_\nu g_{\mu\rho} + \partial_\mu g_{\nu\rho} - \partial_\rho g_{\mu\nu} \right)\,. \tag{A.2}$$

The non-vanishing Christoffel symbols are found to be

$$\Gamma^0_{00} = \alpha\left(\frac{a'}{a}\right)\,, \qquad \Gamma^0_{ij} = \delta_{ij}\left(\frac{a'}{a}\right) a^{2(1-\alpha)}\,, \qquad \Gamma^i_{0j} = \Gamma^i_{j0} = \delta^i_j\left(\frac{a'}{a}\right)\,, \tag{A.3}$$

where primes indicate derivatives with respect to $\eta$. From this, the components of the Ricci tensor

$$R_{\mu\nu} = \partial_\rho \Gamma^\rho_{\mu\nu} - \partial_\nu \Gamma^\rho_{\mu\rho} + \Gamma^\rho_{\mu\nu}\Gamma^\sigma_{\rho\sigma} - \Gamma^\rho_{\mu\sigma}\Gamma^\sigma_{\nu\rho} \tag{A.4}$$

are found to be

$$R_{00} = \frac{3}{a^{2\alpha}}\left[\frac{a''}{a} - \alpha\left(\frac{a'}{a}\right)^2\right] g_{00}\,, \qquad R_{ij} = \frac{1}{a^{2\alpha}}\left[\frac{a''}{a} + (2-\alpha)\left(\frac{a'}{a}\right)^2\right] g_{ij}\,. \tag{A.5}$$

From this, the Ricci scalar $R = g^{\mu\nu}R_{\mu\nu}$ can be computed directly. We find

$$R = \frac{6}{a^{2\alpha}}\left[\frac{a''}{a} + (1-\alpha)\left(\frac{a'}{a}\right)^2\right]\,. \tag{A.6}$$

# B   Energy-momentum tensor of a non-minimally coupled scalar

The energy momentum tensor for the non-minimally coupled $\phi$ is defined in Eq. (9) as

$$T^{\phi}_{\mu\nu} = -\frac{2}{\sqrt{-g}}\frac{\delta(\sqrt{-g}\mathcal{L}_{\phi})}{\delta g^{\mu\nu}} = -2\frac{\delta\mathcal{L}_{\phi}}{\delta g^{\mu\nu}} + g_{\mu\nu}\mathcal{L}_{\phi}\,. \tag{B.1}$$

The remaining variation reads

$$\frac{\delta\mathcal{L}_{\phi}}{\delta g^{\mu\nu}} = -\frac{1}{2}\partial_{\mu}\phi\,\partial_{\nu}\phi - \frac{1}{2}\xi\frac{\delta(\phi^2 R)}{\delta g^{\mu\nu}}\,. \tag{B.2}$$

A useful identity one can show by explicit computation from the expression of $R$ in terms of the metric is the following

$$\frac{\delta(fR)}{\delta g^{\mu\nu}} = fR_{\mu\nu} + (g_{\mu\nu}\nabla^{\sigma}\nabla_{\sigma} - \nabla_{\mu}\nabla_{\nu})f\,, \tag{B.3}$$

for any scalar function $f$ and $\nabla_{\nu}$ is the covariant derivative associated to $g_{\mu\nu}$. Applying this to Eq. (B.2), we obtain

$$\frac{\delta\mathcal{L}_{\phi}}{\delta g^{\mu\nu}} = -\frac{1}{2}\partial_{\mu}\phi\,\partial_{\nu}\phi - \frac{1}{2}\xi\phi^2 R_{\mu\nu} - \frac{1}{2}\xi(g_{\mu\nu}\nabla^{\sigma}\nabla_{\sigma} - \nabla_{\mu}\nabla_{\nu})\phi^2\,,$$

and putting it altogether we get

$$T^{\phi}_{\mu\nu} = \partial_{\mu}\phi\,\partial_{\nu}\phi - g_{\mu\nu}\left(\frac{1}{2}g^{\rho\sigma}\partial_{\rho}\phi\,\partial_{\sigma}\phi + V(\phi)\right) + \xi\left(R_{\mu\nu} - \frac{1}{2}Rg_{\mu\nu} + g_{\mu\nu}\Box - \nabla_{\mu}\nabla_{\nu}\right)\phi^2\,, \tag{B.4}$$

where we have defined $\Box \equiv \nabla^{\sigma}\nabla_{\sigma} = g^{\rho\sigma}\nabla_{\rho}\nabla_{\sigma}$. We would like to compute the trace of this energy-momentum tensor for $\phi$. For a moment, let us write down the result working in $d+1$ dimensions. Then, we find

$$T_{\phi} = (1-d)\frac{1}{2}\partial^{\mu}\phi\,\partial_{\mu}\phi - (d+1)V + \xi G\phi^2 + d\,\xi\Box\phi^2\,, \tag{B.5}$$

where $G = g^{\mu\nu}G_{\mu\nu} = (1-d)R/2$ is the trace of the Einstein tensor with respect to the metric. This expression can be further simplified using $\Box\phi^2 = 2\phi\Box\phi + 2\partial^{\mu}\phi\,\partial_{\mu}\phi$ and the equation of motion Eq. (5) for $\phi$ which gives $\phi\Box\phi = \xi R\phi^2 + \phi V_{,\phi}$. This leads to the following expression

$$g^{\mu\nu}T^{\phi}_{\mu\nu} = 2d\left[\xi - \frac{(d-1)}{4d}\right]\left(\partial^{\mu}\phi\,\partial_{\mu}\phi + \xi R\phi^2\right) - V\left[(d+1) - 2\xi d\,\phi\frac{V_{,\phi}}{V}\right]\,. \tag{B.6}$$

Note that the coefficient of $\left(\partial^{\mu}\phi\,\partial_{\mu}\phi + \xi R\phi^2\right)$ vanishes for $\xi = (d-1)/(4d)$, which is indeed the conformal value of $\xi$ in $d+1$ dimensions. Now setting $d=3$, we get

$$g^{\mu\nu}T^{\phi}_{\mu\nu} = (6\xi - 1)\left(\partial^{\mu}\phi\,\partial_{\mu}\phi + \xi R\phi^2\right) - \left(4V - 6\xi\phi V_{,\phi}\right)\,, \tag{B.7}$$

which indeed leads to Eq. (12) for $T_{\phi} = g^{\mu\nu}T^{\phi}_{\mu\nu}$ quoted in the main text. Let us comment on some specific cases of Eq. (B.7) which reproduce known results. Consider the conformal value for $\xi$ in $d+1 = 4$ dimensions, $\xi = 1/6$. Then, we find

$$g^{\mu\nu}T^{\phi}_{\mu\nu} = -\left(4V - \phi V_{,\phi}\right)\,, \tag{B.8}$$

which vanishes identically in the scaleless cases of $V = 0$ or $V \propto \phi^4$, as expected. For a quadratic potential $V = m^2\phi^2/2$, we get

$$g^{\mu\nu}T^{\phi}_{\mu\nu} = -m^2\phi^2\,, \tag{B.9}$$

which reproduces the result of Ref. [73].

# C   Time evolution and low-storage RK methods

In this appendix, we present the implemented 'low-storage' Runge-Kutta (RK) methods . We also write down an explicit algorithm to evolve our system of equations using these methods. We begin by recalling the reader some facts about RK methods, following Ref. [67]. Consider a vector $\vec{x}(t)$ of $M$-variables $\vec{x}(t) = (x_1(t), \ldots, x_M(t))^T$ and a system of first order differential equations of the type

$$\dot{\vec{x}}(t) = \mathcal{K}\left[\vec{x}(t)\right]. \tag{C.1}$$

Then, a RK method of order $s$ is characterized by a one-step iteration of the type

$$\vec{x}_{n+1} = \vec{x}_n + \sum_{i=1}^{s} c_i k^{(i)}, \tag{C.2}$$

with

$$x^{(i)} = \vec{x}_n + \sum_{j=1}^{s} b_{ij} k^{(j)}, \tag{C.3}$$

$$k^{(i)} = \mathcal{K}\left[\vec{x}^{(i)}\right], \tag{C.4}$$

$$\sum_{i=1}^{s} c_i = 1, \quad c_i < 1. \tag{C.5}$$

This iteration effectively split the time interval $\delta t$ into $s$ subintervals $\delta t = \sum_{i=1}^{s} c_i \delta t$. Note also that, after having introduced conjugate momenta, this is precisely the type of equations we are dealing with. These methods are often represented in terms of *Butcher tableaux*

$$\begin{array}{|cccc|}
\hline
b_{11} & b_{12} & \cdots & b_{1s} \\
b_{21} & b_{22} & \cdots & b_{2s} \\
\ddots & \ddots & \cdots & \ddots \\
b_{s1} & b_{s2} & \cdots & b_{ss} \\
\hline
c_1 & c_2 & \cdots & c_s \\
\hline
\end{array} \tag{C.6}$$

Explicit RK methods have the property $b_{ij} = 0$ for all $j \geq i$. Well known methods of order two are the modified Euler-method (RK2ME) and the midpoint method (RK2MP). For comparison we also show the Butcher tableau of the widely used RK4 algorithm below:

$$\text{RK2ME}: \begin{array}{|cc|}
\hline
0 & 0 \\
1 & 0 \\
\hline
1/2 & 1/2 \\
\hline
\end{array} \qquad \text{RK2MP}: \begin{array}{|cc|}
\hline
0 & 0 \\
1/2 & 0 \\
\hline
0 & 1 \\
\hline
\end{array} \qquad \text{RK4}: \begin{array}{|cccc|}
\hline
0 & 0 & 0 & 0 \\
1/2 & 0 & 0 & 0 \\
0 & 1/2 & 0 & 0 \\
0 & 0 & 1 & 0 \\
\hline
1/6 & 1/3 & 1/3 & 1/6 \\
\hline
\end{array} \tag{C.7}$$

In cases where the limiting factor is memory, such as when solving a system of partial differential equations on large lattices, the memory cost of using higher-order RK methods can become prohibitive. Indeed, generically, one needs to storelve (almost) all of the $k^{(i)}$ coefficients. For a method with $s$-stages, the required additional memory is analogous to simulating $s$ new fields *per field and momentum*.

Interestingly, there exists a subclass of RK methods which eludes this memory requirement; they are referred to as 'low-storage' RK methods [68] (see also Ref. [69] for a recent application in lattice QCD). This method hinges on rewriting of Eqs. (C.2) to (C.4) as

$$\vec{x}_{n+1} = \vec{y}^{(s)}, \tag{C.8}$$

with

$$\vec{y}^{(0)} = \vec{x}_n \,, \tag{C.9}$$

$$\vec{y}^{(i)} = \vec{y}^{(i-1)} + B_i \Delta \vec{y}^{(i)} \,, \tag{C.10}$$

$$\Delta \vec{y}^{(i)} = A_i \Delta \vec{y}^{(i-1)} + \delta t \, , \mathcal{K} \left[ \vec{y}^{(i-1)} \right] \,, \tag{C.11}$$

with the further requirement $A_1 = 0$. Note that all second-order, and some third-order RK methods can be put in this form; we refer the interested reader to Refs. [68, 69] and those therein for more information. It is easy to see that the second order methods introduced above can be recast in this form using the following coefficients:

$$\text{RK2ME}: \quad
\begin{array}{c|cc}
 & A_i & B_i \\
\hline
 & 0 & 1 \\
 & -1 & 1/2
\end{array}
\qquad
\text{RK2MP}: \quad
\begin{array}{c|cc}
 & A_i & B_i \\
\hline
 & 0 & 1/2 \\
 & -1/2 & 1
\end{array}
\tag{C.12}$$

We have also implemented the following third order method from Ref. [74] which is argued to have desirable stability properties. The coefficients below are the rational form of the ones presented in Section 3. of this reference, for $c_3 = (2 + 10^{1/3})/6$:

$$\text{RK3\_4}: \quad
\begin{array}{c|cc}
 & A_i & B_i \\
\hline
 & 0 & 0.06688758201974097 \\
 & -0.7825460361923583 & 2.876554598956719 \\
 & -2.042914325731225 & 0.5534657361343982 \\
 & -1.799337253940777 & 0.3912730180961791
\end{array}
\tag{C.13}$$

Finally, fourth order $2N$-storage schemes also exist, we refer the interested reader to Refs. [69, 75] for examples.

Before writing down explicitly the $2N$-storage method applied to our problem, we note that the scheme RK3\_4 has the additional property that the third iteration $\vec{y}^{(3)}$ is already at second order accuracy in $\delta t$. At the extra memory cost of saving the previous solution $\vec{x}_n$ in case the update fails, one can then easily turn it into an adaptive time-step RK scheme. As reviewed in Refs. [69, 74], this can be achieved by estimating the distance $\Delta$ in some norm between $\vec{y}^{(3)}$ and $\vec{y}^{(4)}$. If this distance is smaller than some requested tolerance $\epsilon$, the update is accepted; if not it is rejected and the step is repeated. In both situation, the time step is updated to

$$\delta t_{\text{new}} = 0.95 \cdot \left( \frac{\epsilon}{\Delta} \right)^{1/3} \delta t_{\text{old}} \,. \tag{C.14}$$

This updated always decrease $\delta t$ when the time step needs to be repeated and almost always increases it when the error is below the requested tolerance. The factor 0.95 and the power 1/3 are empirical determined based on performance. A practical way to define $\Delta$ is to compute the Euclidean distance between the solutions

$$\Delta = |\vec{y}^{(3)} - \vec{y}^{(4)}| = B_4 |\Delta \vec{y}^{(4)}| \,, \tag{C.15}$$

with $|\vec{y}| = \sqrt{\sum_{i=1}^{L} y_l^2}$ for the $L$-component vector $\vec{y} = (y_1, \dots, y_L)^T$. Note that the efficiency of such an adaptive scheme varies from model to model and needs to be studied on a case-by-case basis.

We are now in a position to present a concrete algorithm to evolve the equations presented in Section 3. For every field, momentum and scale factor, we introduce associated auxiliary

variables: $\Delta\tilde{\phi}, \Delta\tilde{\pi}_{\tilde{\phi}}, \{\Delta\tilde{\varphi}_{\mathrm{m}}\}, \{\Delta\tilde{\pi}_{\tilde{\varphi}_{\mathrm{m}}}\}, \Delta a, \Delta\pi_a$. We can then implement a generic $s$-stage $2N$-storage RK method as follows

$$
\begin{aligned}
&\left.
\begin{aligned}
\tilde{\pi}_{\tilde{\phi}}^{(0)} &\equiv \tilde{\pi}_{\tilde{\phi}}(\mathbf{n}, n_0) \\
\tilde{\phi}^{(0)} &\equiv \tilde{\phi}(\mathbf{n}, n_0) \\
\pi_a^{(0)} &\equiv \pi_a(n_0) \\
a^{(0)} &\equiv a(n_0) \\
\{\tilde{\pi}_{\tilde{\varphi}_{\mathrm{m}}}^{(0)}\} &\equiv \{\tilde{\pi}_{\tilde{\varphi}_{\mathrm{m}}}(\mathbf{n}, n_0)\} \\
\{\tilde{\varphi}_{\mathrm{m}}^{(0)}\} &\equiv \{\tilde{\varphi}_{\mathrm{m}}(\mathbf{n}, n_0)\} \\
\tilde{R}^{(0)} &\equiv \tilde{R}(n_0)
\end{aligned}
\right\}
\implies
\left\{
\begin{aligned}
\Delta\tilde{\pi}_{\tilde{\phi}}^{(p)} &= A_p \Delta\tilde{\pi}_{\tilde{\phi}}^{(p-1)} + \delta\tilde{\eta}\mathcal{K}_{\tilde{\phi}}\left[\tilde{\phi}^{(p-1)}, \{\tilde{\varphi}_{\mathrm{m}}^{(p-1)}\}, \tilde{R}^{(p-1)}\right] \\
\Delta\tilde{\phi}^{(p)} &= A_p \Delta\tilde{\phi}^{(p-1)} + \delta\tilde{\eta}a^{(1)3-\alpha}\tilde{\pi}_{\tilde{\phi}}^{(p-1)} \\
\Delta\pi_a^{(p)} &= A_p \Delta\pi_a^{(p-1)} + \delta\tilde{\eta}\mathcal{K}_a\left[\tilde{R}^{(p-1)}\right] \\
\Delta a^{(p)} &= A_p \Delta a^{(p-1)} + \delta\tilde{\eta}a^{(1)\alpha-1}\pi_a^{(p-1)} \\
\{\Delta\tilde{\pi}_{\tilde{\varphi}_{\mathrm{m}}}^{(p)}\} &= \left\{A_p \Delta\tilde{\pi}_{\tilde{\varphi}_{\mathrm{m}}}^{(p-1)} + \delta\tilde{\eta}\mathcal{K}_{\tilde{\varphi}_{\mathrm{m}}}\left[\tilde{\phi}^{(p-1)}, \{\tilde{\varphi}_{\mathrm{m}}^{(p-1)}\}\right]\right\} \\
\{\Delta\tilde{\varphi}_{\mathrm{m}}^{(p)}\} &= \left\{A_p \Delta\tilde{\varphi}_{\mathrm{m}}^{(p-1)} + \delta\tilde{\eta}\mathcal{D}_{\tilde{\varphi}_{\mathrm{m}}}\left[\tilde{\phi}^{(p-1)}, \{\tilde{\varphi}_{\mathrm{m}}^{(p-1)}\}\right]\right\} \\
\tilde{\pi}_{\tilde{\phi}}^{(p)} &= \tilde{\pi}_{\tilde{\phi}}^{(p-1)} + B_p \Delta\tilde{\pi}_{\tilde{\phi}}^{(p)} \\
\tilde{\phi}^{(p)} &= \tilde{\phi}^{(p-1)} + B_p \Delta\tilde{\phi}^{(p)} \\
\pi_a^{(p)} &= \pi_a^{(p-1)} + B_p \Delta\pi_a^{(p)} \\
a^{(p)} &= a^{(p-1)} + B_p \Delta a^{(p)} \\
\{\tilde{\pi}_{\tilde{\varphi}_{\mathrm{m}}}^{(p)}\} &= \left\{\tilde{\pi}_{\tilde{\varphi}_{\mathrm{m}}}^{(p-1)} + B_p \Delta\tilde{\pi}_{\tilde{\varphi}_{\mathrm{m}}}^{(p)}\right\} \\
\{\tilde{\varphi}_{\mathrm{m}}^{(p)}\} &= \left\{\tilde{\varphi}_{\mathrm{m}}^{(p-1)} + B_p \Delta\tilde{\varphi}_{\mathrm{m}}^{(p)}\right\} \\
\tilde{R}^{(p)} &= \tilde{R}\left[\tilde{\phi}^{(p)}, \tilde{\pi}_{\tilde{\phi}}^{(p)}, \{\tilde{\varphi}_{\mathrm{m}}^{(p)}\}, \{\tilde{\pi}_{\tilde{\varphi}_{\mathrm{m}}}^{(p)}\}\right]
\end{aligned}
\right\}_{p=1,\dots,s}
\end{aligned}
$$

$$
\implies
\left\{
\begin{aligned}
\tilde{\pi}_{\tilde{\phi}}(\mathbf{n}, n_0+1) &= \tilde{\pi}_{\tilde{\phi}}^{(s)} \\
\tilde{\phi}(\mathbf{n}, n_0+1) &= \tilde{\phi}^{(s)} \\
\pi_a(n_0+1) &= \pi_a^{(s)} \\
a(n_0+1) &= a^{(s)} \\
\{\tilde{\pi}_{\tilde{\varphi}_{\mathrm{m}}}(\mathbf{n}, n_0+1)\} &= \left\{\tilde{\pi}_{\tilde{\varphi}_{\mathrm{m}}}^{(s)}\right\} \\
\{\tilde{\varphi}_{\mathrm{m}}(\mathbf{n}, n_0+1)\} &= \left\{\tilde{\varphi}_{\mathrm{m}}^{(s)}\right\} \\
\tilde{R}(n_0+1) &= \tilde{R}^{(s)}
\end{aligned}
\right.
\tag{C.16}
$$

The final piece involves implementing Eq. (36), this can be explicitly checked at every time step and provides a robust way to check the stability of the algorithm. To implement the adaptive time step, one proceed as explained above. In particular, the error $\Delta$ is computed as a sum over all the fields and all lattice points of the type

$$
\Delta = B_4 \sum_{x\in\Lambda} \sum_{f\in\left\{\tilde{\pi}_{\tilde{\phi}}, \tilde{\phi}, \tilde{a}, \tilde{\pi}_{\tilde{a}}, \{\tilde{\varphi}_{\mathrm{m}}^{(p)}\}, \{\tilde{\pi}_{\tilde{\varphi}_{\mathrm{m}}}^{(p)}\}\right\}} |\Delta f(x)|.
\tag{C.17}
$$

# D  Numerical convergence

As detailed in Section 3, we have used Eq. (21) to evolve the scale factor, leaving Eq. (36) to allow us to access the numerical convergence of our system of equations. In Fig. 7 we show the resulting convergence using this Hubble constraint equation. We observe good convergence across all scenarios, however we note that there is a larger drift in the integrator when considering larger values on the NMC, and therefore larger field excursions. This is independent of whether or not there are additional terms in the NMC fields' scalar potential (both the grey and purple lines in the left-hand panel of Fig. 7 show similar convergence).

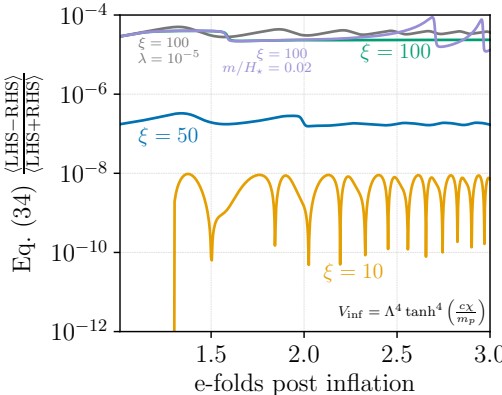
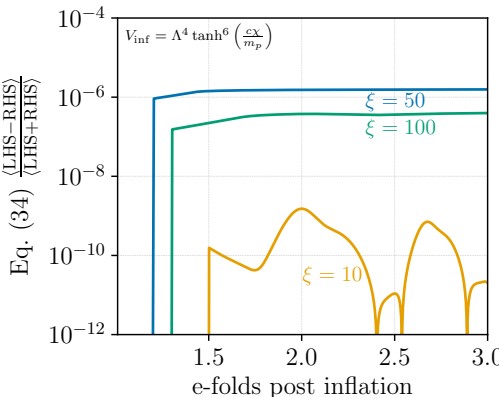

Figure 7: Numerical convergence accessed using the Hubble constraint equation Eq. (36). **Left:** shows the cases we considered for $p = 4$ for the inflationary potential, see Eq. (52). **Right:** same as left, except we consider the steeper potential, $p = 6$.

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
