# Peer review of "Lattice Simulations of Non-minimally Coupled Scalar Fields in the Jordan Frame"

_SciPost Physics, doi:SciPost Phys. 15, 077 (2023)_

## Round 1 · Referee Report · Anonymous (Referee 1) · 2022-12-6

Report

This is the referee report to the manuscript entitled “Lattice Simulations of Non-minimally Coupled Scalar Fields in the Jordan Frame” written by Daniel G. Figueroa, Adrien Florio, Toby Opferkuch and Ben A. Stefanek.

In this manuscript, the authors study numerical lattice simulations of scalar field non-minimally coupled to the gravity in the early universe which is motivated by the renormalization theory in the curved spacetime and the cosmological inflation model.
In particular, the authors newly formulate an algorithm for the lattice simulation in the Jordan frame where the non-minimal coupling is explicitly maintained in the Lagrangian. This is in contrast to most of the previous studies in which the conformal transformation of the metric is performed in order to make the non-minimal coupling term absent in the Lagrangian.

First, the authors have carefully derived the self-contained system of evolution equations for both the non-minimally coupled scalar field and the background expansion of the universe in the Jordan frame. Then, the authors give a formulation for the numerical lattice simulation by discretizing the spacetime variables and replacing spatial derivatives of the scalar field with the finite deferences. The formulation is clearly written in the main text and the technical detail is sufficiently supplemented in the appendices.

In addition, the authors focus on the geometric preheating model as an explicit application of their formulation. Starting from the analysis of the linear evolution of the scalar field sourced by the inflationary fluctuations, which sets the initial values for the lattice simulation, the lattice simulation is performed to follow the subsequent non-linear evolution of the system. Resultant power spectra of the non-minimally coupled scalar field computed by lattice simulations are carefully compared with the results of the linear analysis and the non-linear effect is clearly shown. The evolution of the energy density of the system is computed as well and the authors discuss how the reheating proceeds in this model depending on the non-minimal coupling parameter and the inflaton potential.

Finally, the manuscript contains clear summary and conclusions. The authors also mention their plan to provide publicly available code based on the algorithm presented in the main text. It will allow anyone to reproduce all of the results in this manuscript.

On the whole, the discussion is clear without any ambiguities. In particular, this work opens a new avenue for simulating the non-minimal coupling model and also it will become useful reference especially for users of publicly available CosmoLattice code provided by authors themselves. Thus, I think this work satisfies the acceptance criteria and desrves the publication in SciPost Physics.

---

## Round 1 · Referee Report · Anonymous (Referee 2) · 2022-12-23

Strengths

  1. Carefully articulated formalism---detailed, reproducible, and clear
  2. Validation and comparison of nonlinear simulation results with linearized solutions
  3. Insightful interpretation of the underlying physics in terms of the Jordan-frame field equations
  4. A useful reference to accompany public software

Weaknesses

  1. Inadequate discussion of prior work, chiefly previous numerical simulations of similar models
  2. Lacking motivation and technical justification for the Jordan-frame approach compared to the Einstein frame
  3. New results are not especially novel

Report

The authors present a numerical procedure to perform 3D, nonlinear simulations of scalar field theories nonminimally coupled to gravity. They choose to work in the Jordan frame, where the scalars' kinetic structure remains canonical. They provide a proof-of-concept by simulating geometric preheating after inflation and establishing consistency with linearized calculations (where applicable). Their exposition is clear, well-written, and reproducible and will serve as a useful accompaniment to the software implementation they plan to make publicly available.

The main shortcoming of this paper as a submission to SciPost Physics is its lack of novelty, given that numerous papers have studied very similar models in detail using similar numerical methods. The main technical requirement for this class of models---the use of nonsymplectic integration algorithms---is supported by several other public software packages, even if the specific model the authors consider has not been implemented. In my view, this submission's composition and strengths, as well as the authors' intention to make their own implementation publicly available, make it very well suited for SciPost Physics Codebases and I would recommend publication there after a satisfactory revision.

Requested changes

The main revisions I request are the following:

1) Though the authors' introduction provides a thorough overview of the relevance and motivation for nonminimally coupled scalar fields, the current submission is most lacking in its discussion of its contributions relative to existing literature. In particular, numerous papers have studied numerous variations of the geometric preheating model---e.g., the authors' Refs. [37-40], and in particular, their Ref. [40] and [a] and [b] (which the authors do not currently reference, linked below) perform numerical simulations of highly similar models, generalized to multifield inflation cases where both scalars are nonminimally coupled. Since the authors' aim is to present (and make publicly available) a numerical scheme for solving such systems, a more thorough comparison of their simulation results to those of existing work is warranted.

[a] https://arxiv.org/abs/2005.00433 [b] https://arxiv.org/abs/2007.10978

2) Since the authors advertise the use of the Jordan frame as a positive feature of their scheme, I would have liked to see more substantive discussion of its benefits. While it's convenient to avoid the need to perform the conformal transformation to the Einstein frame, I might expect CosmoLattice (with its symbolic capabilities) to be able to automate that process. As such, it would be valuable to know if the Jordan frame offers other advantages---if, say, the equations are more numerically stable or less computationally expensive. In addition, the authors should note that 3D, nonlinear simulations of nonminimally coupled scalars in the Jordan frame were performed before: [c] considered a preheating of a nonminimally coupled inflaton in the Jordan frame and solved the same equation as the authors' Eqn. 21 (c.f. Eqn 12 in [c]).

[c] https://arxiv.org/abs/1905.13647

In addition, I have the follow more specific/technical questions and comments:

3) Though utilizing the trace of the Einstein equations provides a slick means to specify the background evolution, the dependence of $p_\phi$ (Eqn. 18) on $a''$ is only algebraic. The authors might note that rearranging Eqns. 13 and 14 to isolate derivatives of the scale factor yields a result consistent with Eqn 21.

4) Figure 3, which presents some of the most important results of the simulations (and comparisons to the linear analysis), is not discussed nor even referenced in the main text. This should be amended. I also have two questions regarding it: a) The $N = 2$ lines in the left panel curiously depict an apparent increase in spectral structure in the nonlinear simulations compare to the linearized results. The typical first effect of nonlinearities is to wash out any particular resonance structure that arises in the linear regime. On the other hand, in the linear regime such oscillations in wavenumber could simply be phase offsets due to evaluating the spectra at slightly different times. It would be interesting if the authors could determine whether this discrepancy has a nontrivial cause. b) The simulation results in the right panel exhibit a very broad and flat spectrum out to the UV, but the results appear to be truncated. The authors should display the full spectra to enable evaluating the extent of validity of these simulations. Though I expect their main qualitative point---that self-interactions quench resonance---to be insensitive to resolution effects, the authors should be careful and upfront about assessing convergence (especially so that their submission can serve as a guide for others to properly and effectively use their software).

5) The authors should specify the physical size of the simulation volume and the timestep size used in all simulations.

6) To validate the authors' evolution scheme for FLRW expansion for this class of models, and since Runge-Kutta methods typically incur numerical dissipation error (affecting the satisfaction of conservation laws), the authors should report the performance of their numerical scheme in terms of the degree of violation of the Friedmann constraint (i.e., Eqn. 13).

7) The vertical limits of the left panel of Figure 4 truncate the spikes in the Ricci scalar for large $\xi$. If these indeed take values too large to fit into the axes limits, the authors should quote the peak value in the caption or main body (and, if available, the expected scaling with $\xi$). Likewise for the left panel of Figure 5.

8) It would be helpful if the captions of Figures 4 and 5 specified that the potential $V(\phi) = 0$.

9) Have the authors tested the adaptive time stepping routine they describe in Appendix C? To my knowledge, though use of low-storage Runge-Kutta methods is fairly common, adaptive routines have been little used in 3D simulations and it would be interesting to explore their utility for, e.g., the authors' model. Do the typical choices for timesteps yield, say, percent-level accuracy when compared to results using adaptive stepping? Do the adaptive routines provide substantial savings in simulation runtime? If possible, providing guidance for future users on best practice and potential pitfalls would be valuable (especially for this submission as a SciPost Physics Codebases publication).

10) At the top of page 17, I suspect the phrase "one need to solve (almost) all of the $k^{(i)}$ coefficients" is meant to say "one needs to store", and there appears to be a stray comma in equation C11.

  • validity: high
  • significance: good
  • originality: low
  • clarity: top
  • formatting: excellent
  • grammar: excellent

Author:  Toby Opferkuch  on 2023-02-22  [id 3394]

(in reply to Report 2 on 2022-12-23)

Thank you for your careful reading of our manuscript. Please see attached pdf for our reply to your comments.

Attachment:

referee-reply.pdf

---

## Round 2 · Referee Report · Anonymous (Referee 2) · 2023-3-19

Report

The authors' revision addresses most of my points adequately, but a number of their responses (to points 1-4) still should be added to the text of their submission. In particular, the main text still does not ever reference nor discuss Figure 3. At the least, their response to 4a and 4b should be added to the discussion, to highlight the extent of agreement with linear solutions and explain the physics of backreaction that leads to discrepancies. In addition, if the authors think there are subtleties in extracting physical/frame-invariant results and comparing to results in the Einstein frame (per their response to points 1 and 2), they should raise these issues in, say, their concluding section for the awareness of readers evaluating whether to use the authors' method to study such scenarios (even if detailed investigation is left to future work) .

With these revisions (and those the authors have already made), I would find the manuscript suitable for publication. However, given its limited scope and novelty, I still do not consider the submission to meet the standards of SciPost Physics. My recommendation remains publication in SciPost Physics Codebases (or SciPost Physics Core, if the format is not suitable for Codebases).

---

## Round 2 · Author Response

Dear Editor and Referees,

We would like to thank the referees for the very careful and thorough reading of our manuscript.

Referee report 1:

This is the referee report to the manuscript entitled “Lattice Simulations of Non-minimally Coupled Scalar Fields in the Jordan Frame” written by Daniel G. Figueroa, Adrien Florio, Toby Opferkuch and Ben A. Stefanek.

In this manuscript, the authors study numerical lattice simulations of scalar field non-minimally coupled to the gravity in the early universe which is motivated by the renormalization theory in the curved spacetime and the cosmological inflation model. In particular, the authors newly formulate an algorithm for the lattice simulation in the Jordan frame where the non-minimal coupling is explicitly maintained in the Lagrangian. This is in contrast to most of the previous studies in which the conformal transformation of the metric is performed in order to make the non-minimal coupling term absent in the Lagrangian.

First, the authors have carefully derived the self-contained system of evolution equations for both the non-minimally coupled scalar field and the background expansion of the universe in the Jordan frame. Then, the authors give a formulation for the numerical lattice simulation by discretizing the spacetime variables and replacing spatial derivatives of the scalar field with the finite deferences. The formulation is clearly written in the main text and the technical detail is sufficiently supplemented in the appendices.

In addition, the authors focus on the geometric preheating model as an explicit application of their formulation. Starting from the analysis of the linear evolution of the scalar field sourced by the inflationary fluctuations, which sets the initial values for the lattice simulation, the lattice simulation is performed to follow the subsequent non-linear evolution of the system. Resultant power spectra of the non-minimally coupled scalar field computed by lattice simulations are carefully compared with the results of the linear analysis and the non-linear effect is clearly shown. The evolution of the energy density of the system is computed as well and the authors discuss how the reheating proceeds in this model depending on the non-minimal coupling parameter and the inflaton potential.

Finally, the manuscript contains clear summary and conclusions. The authors also mention their plan to provide publicly available code based on the algorithm presented in the main text. It will allow anyone to reproduce all of the results in this manuscript.

On the whole, the discussion is clear without any ambiguities. In particular, this work opens a new avenue for simulating the non-minimal coupling model and also it will become useful reference especially for users of publicly available CosmoLattice code provided by authors themselves. Thus, I think this work satisfies the acceptance criteria and desrves the publication in SciPost Physics.

We are very grateful to the referee for their very positive comments on our results.

Referee report 2

The main revisions I request are the following:

  1. Though the authors' introduction provides a thorough overview of the relevance and motivation for nonminimally coupled scalar fields, the current submission is most lacking in its discussion of its contributions relative to existing literature. In particular, numerous papers have studied numerous variations of the geometric preheating model---e.g., the authors' Refs. [37-40], and in particular, their Ref. [40] and [a] and [b] (which the authors do not currently reference, linked below) perform numerical simulations of highly similar models, generalized to multifield inflation cases where both scalars are nonminimally coupled. Since the authors' aim is to present (and make publicly available) a numerical scheme for solving such systems, a more thorough comparison of their simulation results to those of existing work is warranted.

    [a] https://arxiv.org/abs/2005.00433

    [b] https://arxiv.org/abs/2007.10978

    We wholeheartedly agree with the referee on the importance and necessity of comparing previous interesting cases presented in the literature, with our proposed Jordan-frame technique. This is however no small endeavor, as making a proper comparison requires significant extensions of our current code. This arises as the models considered in these references are not exactly the same and, in addition, approach solving the dynamics in a different fashion. This leads to two main problems: Firstly, we would need to perform our simulations in the Einstein frame to ensure that our initial conditions are the same as it is currently unclear that a comparison between an Einstein-frame and Jordan-frame simulation will yield the same result. And secondly, this requirement of an Einstein frame simulation yields the additional challenge of non-canonical kinetic terms due to the presence of more than one non-minimally coupled scalar in the above models. We are planning to do such a comparison with some of the most relevant scenarios in the literature, like e.g.~Refs.[38-41] (+[a]), but this study is still a major undertaking and will (in our opinion) constitute a project unto itself, which we hope to begin in the not too distant future. In the present manuscript we content ourselves with solving (for what we believe to be the first time) the non-linear regime of the original geometric preheating scenario (Ref.~[29]), going beyond the linear regime that was first considered in Refs.~[29,30]. Given the above additional non-trivial hurdles, we hope that the referee can be persuaded that the results in their present form are of sufficient interest for SciPost. Lastly, we have added the two aforementioned references in our introduction.

  2. Since the authors advertise the use of the Jordan frame as a positive feature of their scheme, I would have liked to see more substantive discussion of its benefits. While it's convenient to avoid the need to perform the conformal transformation to the Einstein frame, I might expect CosmoLattice (with its symbolic capabilities) to be able to automate that process. As such, it would be valuable to know if the Jordan frame offers other advantages---if, say, the equations are more numerically stable or less computationally expensive. In addition, the authors should note that 3D, nonlinear simulations of nonminimally coupled scalars in the Jordan frame were performed before: [c] considered a preheating of a nonminimally coupled inflaton in the Jordan frame and solved the same equation as the authors' Eqn. 21 (c.f. Eqn 12 in [c]).

    [c] https://arxiv.org/abs/1905.13647

    We fully agree with the referee on the relevance of a full comparison of the Einstein and Jordan frame, and as matter of fact this is already in our pipeline for future work. A proper comparison is part of the work we referred to in our response to point 1, which will be a project (or even series of projects) of its (their) own, given all the aspects that will be required to analyze. The advantages/disadvantages and similarities/discrepancies, between models studied in the Jordan frame and in the Einstein frame, will constitute undoubtedly a very interesting project which will have the twofold purpose of comparing the efficiency of techniques, and the physics itself. Such proper study falls therefore beyond the scope of the current paper, where we simply content ourselves with demonstrating the ability of our algorithm for solving the non-linear in-homogeneous dynamics.

    We can, in any case, try to address the question of the referee, by making already a few comments about the comparison between working directly in the Jordan frame vs the Einstein frame. The first is that in the Einstein frame, one either has to deal with a non-canonical kinetic term (which is not yet implemented in CosmoLattice, though this is now work in progress at the time of writing) or to perform a derivative field re-definition to canonically normalize it. In the latter case, to obtain the potential in the Einstein frame, one must solve a differential equation for the new field in terms of the old variable and then invert the solution, which adds computational complexity. Additionally, it is not clear that the kinetic term is always diagonalizable when considering multiple non-minimally coupled scalar fields. Therefore, CosmoLattice is not currently at the point where the transformation to the Einstein frame can be automated in all cases, which likely requires support for non-minimal kinetic terms. In any case, working directly in the Jordan frame trivially avoids these potential issues, and once we implement the required technical aspects to solve the dynamics in the Einstein frame, we'll be able to make explicit assessments on the comparison about stability, computational cost, advantages/disadvantages, etc, between solving the dynamics in one frame versus the other.

    Finally, we also thank the referee for bringing [c] to our attention. Our more general Eqn. 21, which is valid for an arbitrary potential and time variable, reduces to their Eqn. 12 in the case of a quartic potential and in conformal time. We have added a footnote on pg. 5 stating these points with a citation to [c]. In any case, we note that our Eqn. 21 simply gives an expedient way to evolve the background. One could also integrate the second Friedmann equation by direct substitution of the pressure and energy density, and the result would be the same.

  3. Though utilizing the trace of the Einstein equations provides a slick means to specify the background evolution, the dependence of $p_\phi$ (Eqn. 18) on $a''$ is only algebraic. The authors might note that rearranging Eqns. 13 and 14 to isolate derivatives of the scale factor yields a result consistent with Eqn 21.

    Indeed, we agree that they are consistent, as such re-arrangement leads to an expression proportional to $\rho - 3p$, which is the trace of the energy-momentum tensor.

  4. Figure 3, which presents some of the most important results of the simulations (and comparisons to the linear analysis), is not discussed nor even referenced in the main text. This should be amended. I also have two questions regarding it:

    1. The $N=2$ lines in the left panel curiously depict an apparent increase in spectral structure in the nonlinear simulations compare to the linearized results. The typical first effect of nonlinearities is to wash out any particular resonance structure that arises in the linear regime. On the other hand, in the linear regime such oscillations in wavenumber could simply be phase offsets due to evaluating the spectra at slightly different times. It would be interesting if the authors could determine whether this discrepancy has a nontrivial cause.

    The resulting lattice power spectrum compared to the linear one must be different as for the linear analysis we do not include the backreaction of the NMC field in the Friedmann equation. We see at $N\sim1.75$ the energy densities of the inflaton and the spectator fields become similar signaling the start of the back-reaction (see the blue lines in Fig.~(4)). At $N=2$ differences in both the peak and also the UV tail of the spectrum arise. The origin of the additional structure in the peak of the lattice results arises from the Ricci scalar remaining positive once the backreaction occurs. Subsequently the NMC just behaves as a free oscillator and is no longer driven.

    2. The simulation results in the right panel exhibit a very broad and flat spectrum out to the UV, but the results appear to be truncated. The authors should display the full spectra to enable evaluating the extent of validity of these simulations. Though I expect their main qualitative point---that self-interactions quench resonance---to be insensitive to resolution effects, the authors should be careful and upfront about assessing convergence (especially so that their submission can serve as a guide for others to properly and effectively use their software).

    We thank the referee for pointing out this coverage issue. It is true that because of the quartic coupling re-scattering of modes leads to additional power in the UV spectrum. We have re-run the simulation covering better the UV scales while maintaining comparable IR coverage (see figure below). We indeed find that this resolves much better the expected fall-off of the UV tail. However, the spectral peak amplitude does not change and therefore neither does the expectation value of the field. We emphasize that re-running the simulation for this case was only necessary due to the effects of the quartic coupling (grey line of fig. 6 [top-right] has changed slightly due to UV oscillations), all other results in the manuscript remain unchanged. We have updated the lattice parameters used in the manuscript to reflect these changes.

  5. The authors should specify the physical size of the simulation volume and the timestep size used in all simulations.

    In all lattice simulations with $\lambda=0$ we have used $N = 240$ and $k_\text{IR} = 4\times 10^{-3} H_i$ (except for $\xi = 10$ where $k_\text{IR} = 2.5\times10^{-3} H_i$) while the time-step has been chosen as $H_i \delta t=0.01$. For $\lambda=10^{-5}$, we used $N=512$, $k_\text{IR} = 10^{-2} H_i$. This choice allows for good coverage of the spectrum where tachyonic growth occurs for a number of e-folds after inflation ends. We have updated the manuscript accordingly, see the paragraph underneath Eq. (57).

  6. To validate the authors' evolution scheme for FLRW expansion for this class of models, and since Runge-Kutta methods typically incur numerical dissipation error (affecting the satisfaction of conservation laws), the authors should report the performance of their numerical scheme in terms of the degree of violation of the Friedmann constraint (i.e., Eqn. 13).

    In a new appendix we show the energy conservation for all cases considered in our manuscript. As described in the text we utilize Eq. (36) as a cross-check of the numerical convergence while we have used Eq. (21) to evolve the scale factor.

  7. The vertical limits of the left panel of Figure 4 truncate the spikes in the Ricci scalar for large $\xi.$ If these indeed take values too large to fit into the axes limits, the authors should quote the peak value in the caption or main body (and, if available, the expected scaling with $\xi$). Likewise for the left panel of Figure 5.

    For Fig.~(4) LHS, the two peaks extend to values of $R/H^2 = 20.8\, (49.1)$ for $\xi = 50\, (100)$, respectively. While for Fig.~(5) RHS the $w$ peaks extends to $w=-2\,(-5.1)$ for $\xi=50\,(100)$. We have added these values to the figure captions in the manuscript.

  8. It would be helpful if the captions of Figures 4 and 5 specified that the potential $V(\phi)=0$.

    Done!

  9. Have the authors tested the adaptive time stepping routine they describe in Appendix C? To my knowledge, though use of low-storage Runge-Kutta methods is fairly common, adaptive routines have been little used in 3D simulations and it would be interesting to explore their utility for, e.g., the authors' model. Do the typical choices for timesteps yield, say, percent-level accuracy when compared to results using adaptive stepping? Do the adaptive routines provide substantial savings in simulation runtime? If possible, providing guidance for future users on best practice and potential pitfalls would be valuable (especially for this submission as a SciPost Physics Codebases publication).

    We did implement the adaptive routine described in Appendix C. We agree with the referee of the interest of these adaptive routines and that a dedicated study of performance is warranted. While some performance gain is expected across a range of different models and parameter space (which we observed on some small test simulations of some simple power-law inflationary model), the dynamics of the model in this work is so fast that the overhead of the adaptive routine wins over the potential gain. As a result, we decided to defer any such study to further, more appropriate works. We still decided to include it as a comment in the appendix here as this is an interesting and straightforward application of the low-storage algorithms presented. We added a sentence at the end of the relevant paragraph to make the reader aware that a performance gain is not guaranteed: "Note that the efficiency of such an adaptive scheme varies from model to model and needs to be studied on a case-by-case basis."

10. At the top of page 17, I suspect the phrase "one need to solve (almost) all of the $k(i)$ coefficients" is meant to say "one needs to store", and there appears to be a stray comma in equation C11.

Indeed, corrected!

---

## Round 3 · Author Response

Dear Editor and Referees,

Thank you for the positive comments. Please see the list of changes made to the manuscript to better incorporate the previous reply to the referees. These changes include an expanded introduction, a more detailed discussion of the results (including reference to figure 3), and finally inclusion of the frame equivalence issues in the case of initial conditions driven by quantum fluctuations.

---

## Round 3 · List of Changes

• Lengthened the second last paragraph of the introduction (bottom of page 2), to explain cases where the conformal mapping between the Jordan and Einstein frames is not valid.
  • Lengthened the paragraph beginning at the end of page 11 discussing the results of the geometric preheating model. Here a thorough discussion of the power spectrum is included.
  • Added a paragraph at the bottom of page 13 beginning with``Before turning to the evolution''. Here we discuss the role of the quartic in modifying the resulting power spectrum.
  • Significant modifications of the first and the final paragraph of the introduction to better include a discussion of the frame equivalence.

---

## Editorial Decision

published